# Checkpoint kinase 1 is essential for fetal and adult hematopoiesis

Fabian Schuler[1] (ID), Sehar Afreen[2,3], Claudia Manzl[4], Georg Häcker[5] (ID), Miriam Erlacher[2,6,7] & Andreas Villunger[1,8,9,*] (ID)

## Abstract

Checkpoint kinase 1 (CHK1) is critical for S-phase fidelity and preventing premature mitotic entry in the presence of DNA damage. Tumor cells have developed a strong dependence on CHK1 for survival, and hence, this kinase has developed into a promising drug target. *Chk1* deficiency in mice results in blastocyst death due to G2/M checkpoint failure showing that it is an essential gene and may be difficult to target therapeutically. Here, we show that chemical inhibition of CHK1 kills murine and human hematopoietic stem and progenitor cells (HSPCs) by the induction of BCL2-regulated apoptosis. Cell death in HSPCs is independent of p53 but requires the BH3-only proteins BIM, PUMA, and NOXA. Moreover, *Chk1* is essential for definitive hematopoiesis in the embryo. Noteworthy, cell death inhibition in HSPCs cannot restore blood cell formation as HSPCs lacking CHK1 accumulate DNA damage and stop dividing. Moreover, conditional deletion of *Chk1* in hematopoietic cells of adult mice selects for blood cells retaining CHK1, suggesting an essential role in maintaining functional hematopoiesis. Our findings establish a previously unrecognized role for CHK1 in establishing and maintaining hematopoiesis.

**Keywords** apoptosis; BCL2; CHK1; DNA damage; hematopoietic stem cells
**Subject Categories** Autophagy & Cell Death; Cell Cycle; Stem Cells

## Introduction

Long-term HSCs (LT-HSCs) in the adult have self-renewal capacity but reside in a quiescent state for most of their lifetime [1]. LT-HSCs divide asymmetrically into one daughter with short-term (ST) reconstitution potential, differentiating into a multi-potent progenitor (MPP) cell and the other daughter remaining a *bona fide* stem cell [2,3]. MPPs then commit to the myeloid, lymphoid, or erythroid/megakaryocyte lineage. These transient amplifying cells with limited lineage potential provide the organism with all blood cells needed. To fulfill this task over a lifetime, cell cycle entry and quiescence of LT-HSCs and their immediate progeny are tightly regulated, e.g., intrinsically by the polycomb-protein BMI1 and the p53 tumor suppressor [1,4] but they also response to trophic signals from the bone marrow micro-environment in the stem cell niche [5–7], as well as to systemic cues, elicited in response to viral or microbial infections, most notably interferons [8,9].

The serine/threonine kinase checkpoint kinase 1 (CHK1) is a critical cell cycle regulator that controls normal proliferation and is activated in response to DNA damage, thereby also controlling p53 function [10,11]. Especially upon single-stranded DNA breaks, ataxia-telangiectasia and Rad3-related protein (ATR) phosphorylates CHK1, leading to its activation and stabilization [12]. On the one hand, active CHK1 arrests the cell cycle via inhibition of CDC25 phosphatases that are essential for the activity of Cyclin/CDK complexes. CHK1-phosphorylated CDC25A is marked for ubiquitination and therefore proteasomal degradation leading to a drop in CDK2 activity and subsequent G1/S arrest [13,14]. On the other hand, CDC25C is retained in the cytoplasm by 14-3-3 proteins when phosphorylated by CHK1 upon DNA damage, restraining CDK1 activity leading to a G2/M arrest [15]. Moreover, CHK1 promotes the activity of MYT1 and WEE1 kinases that both inhibit CDK1 by phosphorylation, blocking transition from G2 to M-phase [16,17]. Under these conditions, CHK1 can stabilize p53 by direct phosphorylation to tighten cell cycle arrest [18,19]. In the absence of p53, however, cells become highly dependent on CHK1 for cell cycle control, arrest, and repair of DNA damage [12,14], generating a vulnerability that is currently explored as a means to treat cancers with CHK1 inhibitors [11,20].

*Chk1* deletion in mice was shown to be embryonic lethal around E5.5 due to G2/M checkpoint failure. Blastocysts lacking *Chk1*

1   Division of Developmental Immunology, Biocenter, Medical University of Innsbruck, Innsbruck, Austria
2   Division of Pediatric Hematology and Oncology, Department of Pediatrics and Adolescent Medicine, Faculty of Medicine, University of Freiburg, Freiburg, Germany
3   Faculty of Biology, University of Freiburg, Freiburg, Germany
4   Institute of Pathology, Neuropathology and Molecular pathology, Medical University of Innsbruck, Innsbruck, Austria
5   Institute of Medical Microbiology and Hygiene, University Medical Center Freiburg, Freiburg, Germany
6   German Cancer Consortium (DKTK), Freiburg, Germany
7   German Cancer Research Center (DKFZ), Heidelberg, Germany
8   CeMM Research Center for Molecular Medicine of the Austrian Academy of Sciences, Vienna, Austria
9   Ludwig Boltzmann Institute for Rare and Undiagnosed Diseases, Vienna, Austria
    *Corresponding author. Tel: +43-512-9003-70380; Fax: +43-512-9003-73960; E-mail: andreas.villunger@i-med.ac.at

exhibit massive DNA damage and cell death that could not be overcome by co-deletion of *p53* [21,22]. All of this supports the essential function of *Chk1* in cell cycle regulation and the DNA damage response to avoid mutational spread and genomic instability. Of note, a certain percentage of $Chk1^{+/-}$ mice was reported to develop anemia with age, suggesting critical dose-dependent roles in erythropoiesis [23] while conditional deletion of *Chk1* in B and T cells was shown to arrest their development at early proliferative stages due to accumulation of DNA damage and increased cell death [24,25]. This suggests that blood cancer treatment with chemical inhibitors targeting CHK1 may cause collateral damage within the healthy hematopoietic system, at least in cycling lymphoid or erythroid progenitors, yet the role of *Chk1* in early hematopoiesis and stem cell dynamics as well as for adult blood cell homeostasis remains unexplored.

It was reported that *Chk1* mRNA is expressed at significant levels in HSC [23] despite the fact that HSC remain quiescent for the majority of their lifetime. Given the fact that HSC accumulate DNA damage when exiting dormancy [26,27], e.g., under pathological conditions such as substantial blood loss or in response to infection [8,9,28], as well as during natural aging [29,30], it appears appropriate that HSCs arm themselves with CHK1 to immediately deal with the dangers of a sudden proliferative challenge to avoid mutational spread. Yet, another study found that *Chk1* mRNA levels are low in HSC but increase during proliferation-coupled self-renewal or differentiation, along with other DNA damage response and repair genes [29]. Consistent with a direct link to proliferation, *Chk1* mRNA was also found to be higher in fetal liver vs. bone marrow-derived HSC from young or aged mice [29]. Notably here, HSC from old mice suffer from enhanced replication stress upon mobilization that triggers substantial CHK1 activation [30]. Together, this suggests critical roles for CHK1 in stem cell dynamics and prompted us to elucidate its role in the development and survival of hematopoietic stem and progenitor cells. Using a conditional allele for *Chk1* in combination with a CRE-deleter strain allowing recombination of this allele within the hematopoietic system, we were able to highlight the importance of *Chk1* for the establishment of hematopoiesis in the embryo and the survival of HSPCs from mice and men.

## Results

### CHK1 inhibition limits the survival of hematopoietic progenitors by inducing mitochondrial apoptosis

To explore the consequences of CHK1 inhibition on cycling multipotent hematopoietic precursors, we first assessed the response of Hoxb8-immortalized FLT3-dependent progenitor cells (Hoxb8-FL cells). These cells can be generated from murine bone marrow and are multi-potent progenitor (MPP)-like cells that can be differentiated *in vitro* into myeloid and lymphoid lineages [31]. Treatment of Hoxb8-FL cells with two specific CHK1 inhibitors (CHK1i), PF-477736 (PF) or CHIR-124 (CHIR), resulted in time- and dose-dependent cell death, as assessed by propidium iodide (PI) staining of DNA content by flow cytometric analysis (Figs 1A and EV1A, left panel). Hoxb8-FL cells generated from *Vav-BCL2*-transgenic mice or mice lacking the key-apoptotic effectors *Bax* and *Bak* were protected

from cell death, indicating initiation of caspase-dependent mitochondrial apoptosis after CHK1i treatment (Figs 1A and EV1A). Consistently, addition of the pan-caspase inhibitor QVD prevented CHK1i-induced cell death (Fig 1B). The rapid cell death seen in wild-type cells was accompanied by γH2A.X accumulation, p53-stabilization, and PARP1 cleavage, as detected in western analyses (Fig 1C). As γH2A.X phosphorylation is also induced during apoptosis due to CAD-mediated cleavage of DNA between nucleosomes [32], we wondered if this may be a secondary consequence of caspase activation. Consistently, upon CHK1 inhibition, Hoxb8-FL cells lacking BAX/BAK also stabilized p53 and showed significantly lower levels of γH2A.X phosphorylation but no signs of PARP1 cleavage or cell death (Fig 1C). This suggests that in wild-type cells CHK1i treatment causes DNA damage, triggering a p53 response that precedes BCL2-regulated mitochondrial cell death involving caspase activation. Remarkably though, Hoxb8-FL cells derived from the bone marrow of $p53^{-/-}$ mice were still highly susceptible to cell death indicating a minor contribution of p53 target genes to apoptosis upon CHK1 inhibition (Figs 1A and EV1A, left panel). To obtain first mechanistic insight into this p53-independent cell death, we tested a series of Hoxb8-FL cells derived from mice lacking different pro-apoptotic "BH3-only" proteins that act as inducers of BAX and BAK activation in mitochondrial cell death [33,34]. Of note, loss of the BH3-only protein BIM provided some degree of protection when PUMA or NOXA was co-depleted (Fig EV1B and C). Together, this suggests that the BH3-only protein BIM, together with PUMA and NOXA mediate BAX/BAK activation upon CHK1 inhibition.

Interestingly, we also noted an altered cell cycle distribution in cells that failed to undergo cell death upon CHK1 inhibition. The majority of BCL2-overexpressing or BAX/BAK-deficient cells accumulated in the G1 fraction with a near complete loss of S-phase cells, indicating cell cycle arrest at the G1/S boundary when CHK1 function was inhibited and cell death blocked at the same time (Figs 1A and EV1A, right panel). Consistently, BAX/BAK-deficient cells showed a significant induction of *p21* mRNA in response to CHK1 inhibition (Fig 1D). To corroborate this finding, we also performed population-doubling experiments in the presence or absence of CHK1 inhibitor. Indeed, cells overexpressing BCL2 or cells lacking BAX/BAK failed to proliferate when treated with CHK1i and showed near-constant cell numbers over time (Fig 1E).

Together, these findings underscore the importance of CHK1 for hematopoietic progenitor cell proliferation and survival by preventing BAX/BAK-mediated apoptotic cell death. Apoptosis seems to be initiated in response to DNA damage upon CHK1 inhibition but does not require p53. When cell death is blocked, these cells show signs of DNA damage and arrest the cell cycle at the G1/S boundary.

### CHK1 inhibition kills primary mouse and human HSPCs via BCL2-regulated apoptosis

To get a first impression on how CHK1 inhibition affects primary HSPCs, we isolated Lin⁻ Sca1⁺ cKit⁺ (LSK) cells, containing HSCs and MPPs, from the fetal liver of WT or *Vav-BCL2* transgenic embryos or the bone marrow of adult mice. Sorted LSK cells from either source showed only modest cell death when treated with CHK1i *in vitro* but the cell death seen was blocked when BCL2 was overexpressed (Fig 2A, Appendix Fig S1A and B). This suggests that, compared to Hoxb8-FL cells, reduced proliferation rates in

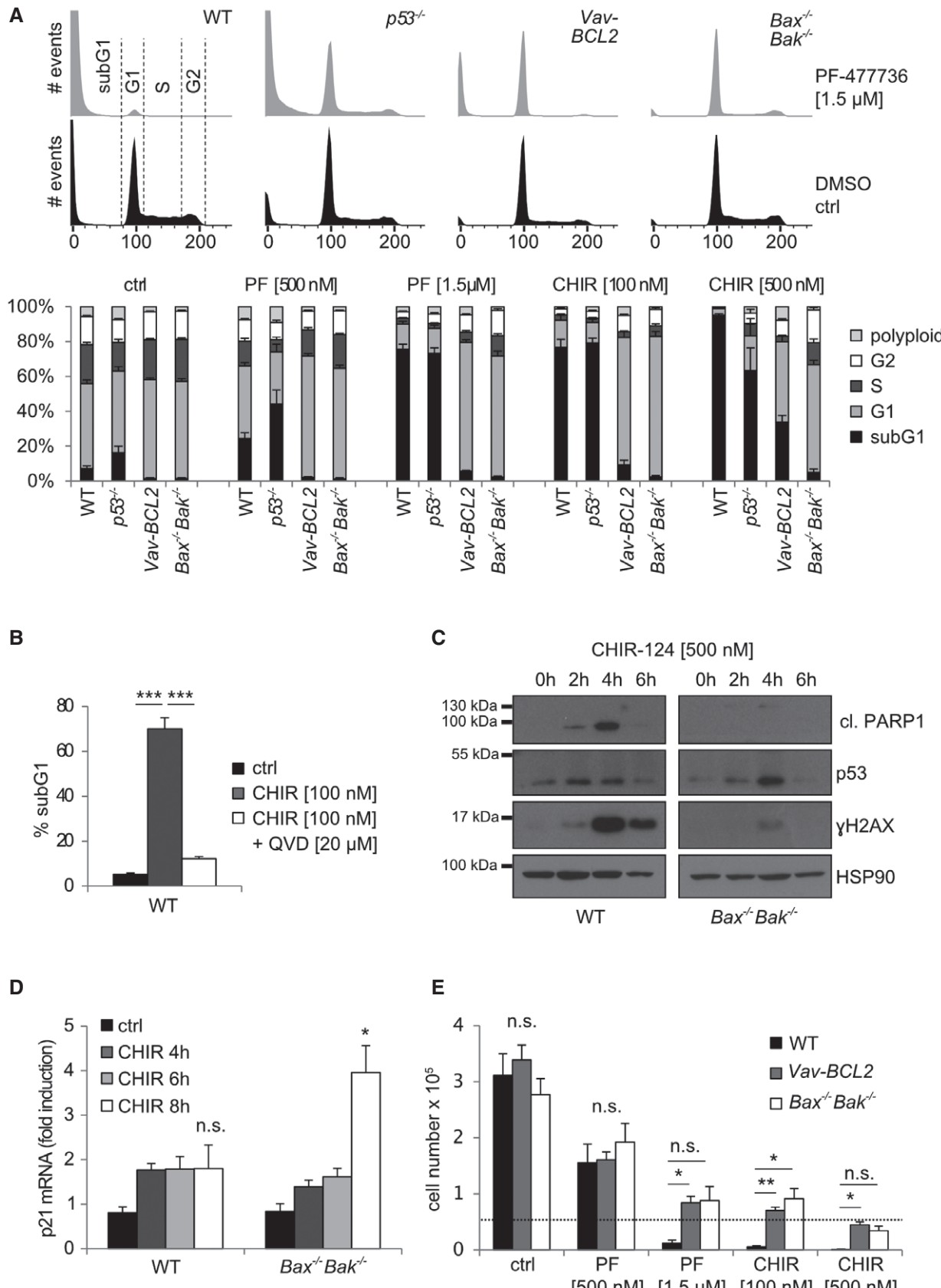

**Figure 1.**

**Figure 1. CHK1 is an essential survival factor for Hoxb8-FL cells.**

A   Hoxb8-FL cells generated from bone marrow of mice of the indicated genotypes were treated for 24 h with graded doses of the CHK1 inhibitors PF-477736 (PF) or CHIR-124 (CHIR). Representative histograms of CHK1i-treated (gray) or DMSO-treated (black) cells analyzed for cell cycle distribution and cell death using Nicoletti staining and flow cytometry are shown.

B   WT Hoxb8-FL cells were treated with CHIR-124 in the presence or absence of the pan-caspase inhibitor QVD. Cell death was assessed after 24 h using flow cytometry as in (A).

C   Western blot of CHIR-treated Hoxb8-FL WT and $Bax^{-/-}Bak^{-/-}$ cells using the indicated antibodies.

D   WT and $Bax^{-/-}Bak^{-/-}$ Hoxb8-FL cells were treated for up to 8 h with 500 nM CHIR, and *p21* mRNA levels were compared to *Hprt* in response to CHK1 inhibition by real-time RT–qPCR. Bars represent means ± s.d. of independent experiments (n = 4–6), performed in duplicates.

E   50,000 Hoxb8-FL cells of the indicated genotypes were seeded to assess cell growth in response to CHK1-inhibition. Cells were counted 48 h postseeding.

Data information: Asterisks indicate significant differences: *$P < 0.05$, **$P < 0.01$, ***$P < 0.001$ using unpaired Student's *t*-test. Bars in (B) represent means ± s.e.m. (n = 2 biological and technical replicates per genotype). Bars in (E) represent means ± s.e.m. (n = 3 biological replicates per genotype).

Source data are available online for this figure.

primary cells may impact on CHK1i sensitivity. Consistently, when we treated total fetal liver (FL) or total bone marrow cell cultures that both contain a substantial number of lineage-committed cycling hematopoietic progenitors with CHK1i, we observed a clear dose-dependent response, driving a significant portion of cells into apoptosis. Again, cell death was significantly reduced when BCL2 was overexpressed (Fig 2B and C). This indicates that actively cycling hematopoietic progenitors are more vulnerable to CHK1i, compared to LSK cells, suggesting that cell cycle rates define susceptibility to CHK1i.

To test whether these observations also hold true for human cells, we purified CD34$^+$ HSPCs from the cord blood that reportedly express low-level CHK1 [35]. Indeed, a significant fraction of CD34$^+$ cells died when treated with increasing doses of CHK1i in the presence of proliferation-inducing cytokines (Fig EV2A). This cell death was potently blocked by the pan-caspase inhibitor QVD, pointing toward mitochondrial cell death (Fig 3A). In line with our findings in mouse cells, cell death was reduced when BCL2 was introduced into CD34$^+$ cells by viral gene transfer (Fig 3B) and led to an enrichment of GFP$^+$ cells upon treatment, marking successful transduction (Figs 3C and EV2B). Moreover, human CD34$^+$ HSPCs lost their potential to form colonies in methyl cellulose when CHK1 was inhibited chemically (Fig EV2C). In line with our experiments using Hoxb8-FL cells, BCL2 overexpression failed to rescue colony formation upon CHK1 inhibition (Fig 3D).

Together, this suggests that BCL2-regulated and caspase-dependent apoptosis is initiated also in CD34$^+$ human HSPCs treated with CHK1 inhibitor expanded *ex vivo* and that blocking cell death is insufficient to restore their clonal outgrowth, most likely because of cell cycle arrest induction.

## CHK1 is essential for establishing hematopoiesis in the fetal liver

Our data indicate that CHK1 might be critical for the development of a functional hematopoietic system. To explore this genetically, we crossed mice harboring a conditional allele of *Chk1* with a CRE-deleter strain specific for the hematopoietic system [36]. Deletion of *Chk1* using *Vav*-driven CRE expression never resulted in viable mice at the time of weaning but caused hematopoietic failure and the death of *Chk1$^{fl/-}$ Vav-Cre* embryos *in utero* (Fig 4A and B). The lack of red blood cells in the fetal liver of *Chk1$^{fl/-}$ Vav-Cre* embryos at E13.5 was easily visible histologically (Fig 4C and D), pointing toward impaired erythropoiesis. Indeed, flow cytometric analysis revealed a decreased percentage of pro-erythroblasts (CD71$^+$ cKit$^+$)

as well as erythroid progenitor (CD71$^+$ CD45$^{low}$) cells in the fetal liver of *Chk1$^{fl/-}$ Vav-Cre* embryos and the erythrocytes present in these embryos appeared to be of primitive origin [37], judged on the lack of cKit expression on their cell surface (Fig EV3A and B). Furthermore, we could observe a clear reduction of all developmental stages of erythropoiesis in the fetal liver of *Chk1$^{fl/-}$ Vav-Cre* embryos (Fig 4E).

Of note, we also failed to detect the typical LSK cell population enriched for HSPCs in the fetal liver. Within the Lin$^-$ fraction of cells, we found instead Sca1$^+$ cells with a near 10-fold reduction of cKit expression (LSK$^{low}$ cells), a phenomenon not seen in littermate controls (Fig 5A). Remarkably, the percentage of these LSK$^{low}$ cells was actually increased about 5-fold leading to comparable overall LSK numbers compared to littermate controls (adding up LSK$^{hi}$ and LSK$^{low}$ cells). In contrast, the fraction of Lin$^-$ LK cells, containing MPPs and more committed progenitors, was significantly reduced, both in relative and absolute terms. Although these cells also shifted into the LK$^{low}$ fraction, their absolute number remained significantly decreased when adding up LK$^{hi}$ and LK$^{low}$ cells (Fig 5A). This suggested that LK cells cannot survive or expand properly in the absence of *Chk1*.

To corroborate this phenotype further, we also performed colony formation assays in methyl cellulose. Neither sorted LSK cells nor total FL cells isolated from *Chk1$^{fl/-}$ Vav-Cre* mice showed colony-forming potential (Fig 5B). A more detailed analysis of the LSK cell pool suggested loss of CD34$^-$Flt3$^-$ LT-HSC accompanied by a relative increase in CD34$^+$Flt3$^-$ ST-HSC. MPPs, characterized as CD34$^+$Flt3$^+$, were found unchanged in percentage using this maker combination (Fig 5C).

Together, this indicates that failure in blood cell development and embryonic lethality in *Chk1$^{fl/-}$ Vav-Cre* mice is potentially due to exhaustion of LT-HSC that actively cycle to colonize the fetal liver after migrating in from the AGM (aorta-gonad mesonephros) region [2,38]. Hematopoietic failure may be driven by the loss of ST-HSC, MPPs, or their immediate progeny that, based on our *in vitro* findings above, may undergo apoptosis in the absence of CHK1.

## DNA damage and subsequent cell death trigger compensatory proliferation and stem cell loss in the absence of *Chk1*

Chemical inhibition of CHK1 induces DNA damage in cultured MPP-like cells and activates a p53-independent but mitochondrial cell death-dependent response (Fig 1). To explore whether this is also responsible for the phenotypes seen *in vivo*, we analyzed

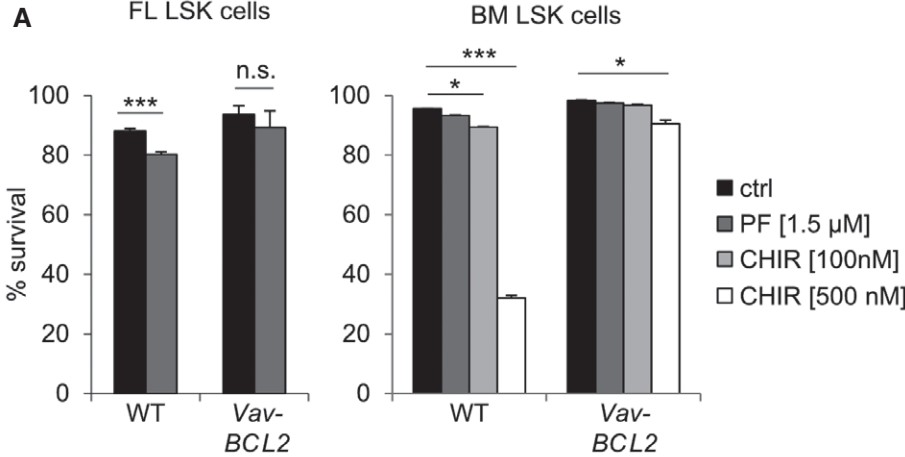

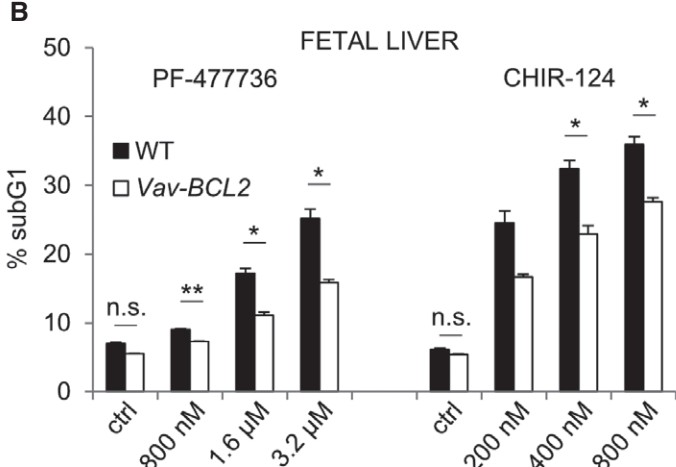

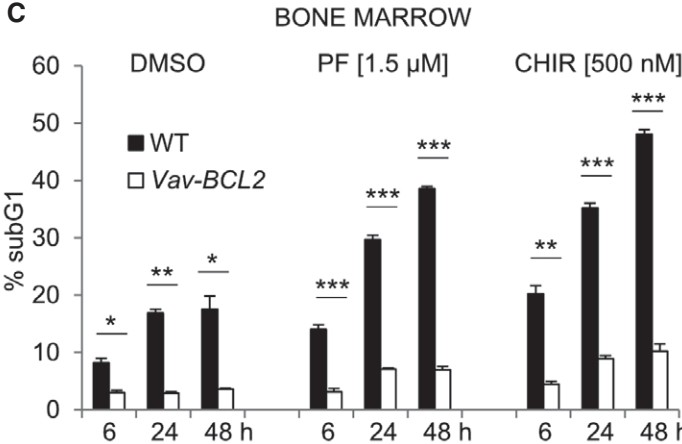

**Figure 2. CHK1 inhibition kills primary murine HSPCs.**

A  LSK (Lin⁻ Sca1⁺ cKit⁺) cells of the indicated genotypes were isolated by cell sorting from the fetal liver at embryonic day E13.5 or the bone marrow of adult mice and treated for 48 h with PF-477736 (1.5 μM). Survival was assessed using Annexin V-staining and flow cytometry. Bars represent means ± s.e.m. from $n = 6$ wild-type and $n = 3$ Vav-BCL2 fetal liver LSK cells (left) and $n = 3$ bone marrow LSK cells (right).

B, C  WT and Vav-BCL2-derived E14.5 total fetal liver cells (B) or total bone marrow cells (C) were treated for 72 h (B) or up to 48 h (C) with different doses of CHK1i. Survival was assessed using Nicoletti staining and flow cytometry. Bars represent means ± s.e.m. ($n = 3$ biological replicates per genotype).

Data information: Asterisks indicate significant differences: *$P < 0.05$, **$P < 0.01$, ***$P < 0.001$ using unpaired Student's *t*-test.

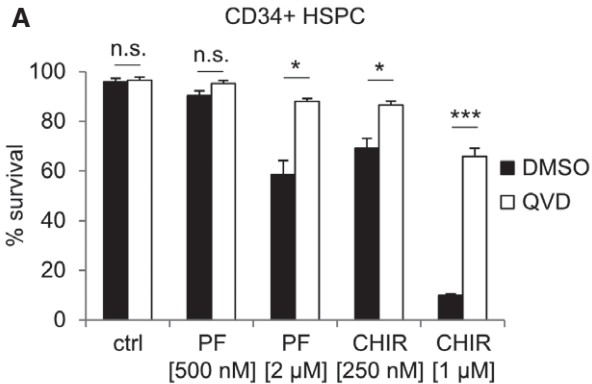

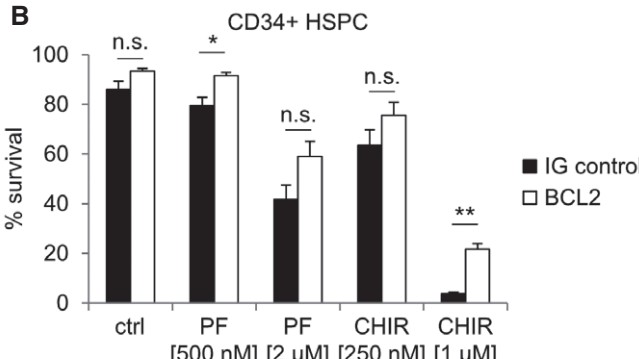

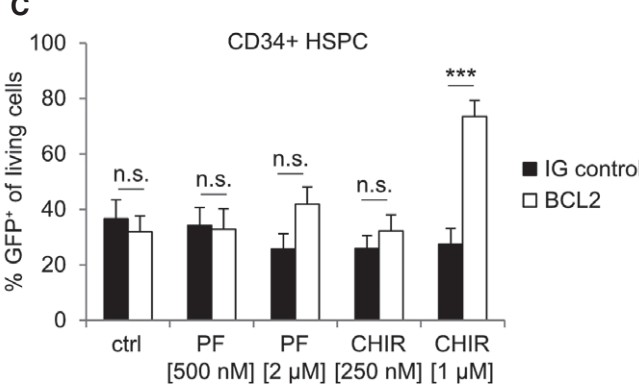

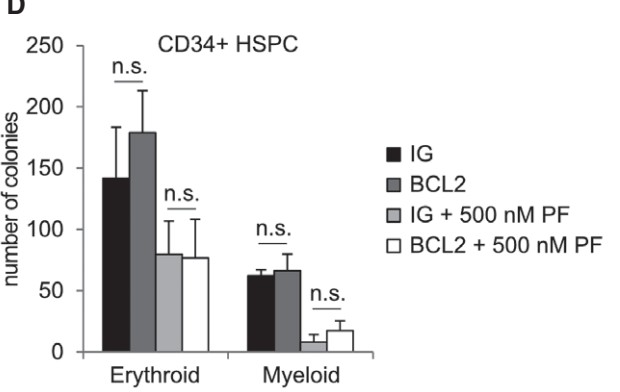

**Figure 3. CHK1 inhibition kills human cord blood-derived CD34⁺ HSPCs in a BCL2 regulated manner.**

A   MACS-purified CD34⁺ human cord blood-derived HSPCs were treated with PF-477736 or CHIR-124 ± the caspase inhibitor QVD (50 μM). Cell death was assessed using Annexin V/7-AAD staining and flow cytometry.

B   CD34⁺ cells were transduced with a BCL2-encoding lentivirus or a control virus, along with IRES-GFP with a MOI of 10 for two consecutive days, followed by CHK1i application for 48 h. Survival was assessed using Nicoletti staining and flow cytometry.

C   Percentage of living GFP⁺ transduced human CD34⁺ HSPCs, cultured in the absence or presence of CHK1i for 48 h.

D   Colony formation potential of human CD34⁺ HSPCs transduced with empty control or BCL2 encoding virus. Colonies were counted 10 days postseeding of 1,000 CD34⁺ cells per plate/sample. "Myeloid colonies" include granulocyte/monocyte, monocyte, granulocyte, and GEMM (granulocyte, erythroid, monocyte/macrophage, and megakaryocyte) colonies.

Data information: Bar graphs in (A–D) represent means ± s.e.m. (independent experiments with four individual cord blood-derived CD34⁺ HSPCs). Asterisks indicate significant differences: *$P < 0.05$, **$P < 0.01$, ***$P < 0.001$ using unpaired Student's *t*-test.

increase in TUNEL-positive cells in E13.5 tissue sections (Fig 6A) and corresponding activation of ATR, detected by western in E13.5 total fetal liver lysates isolated from *Chk1*$^{fl/-}$ *Vav-Cre* embryos (Fig 6B). Moreover, DAPI-negative LSK and LK cells isolated by cell sorting from the fetal liver of *Chk1*$^{fl/-}$ *Vav-Cre* embryos showed a strong increase in γH2A.X phosphorylation (Fig 6C). The increased γH2A.X phosphorylation seen in these embryos provides a rational for the noted stem cell depletion (loss of G0 cells), due to increased proliferation of LSK and LK cells, forced into cycle in an attempt to compensate for the loss of erythroid cells or their progenitors. Consistently, LSK and LK cells lacking CHK1 were devoid of Ki67-negative cells, indicating increased proliferation rates and simultaneously showed increased percentages of sub-G1 cells. Interestingly, the percentage of cells in S- or G2/M-phase was unaltered (Fig 6D and E). Together, this suggests that these cells accumulate DNA damage while cycling in the absence of CHK1, leading to apoptosis, which could explain the loss of LK cells seen. The increased death of LK cells appears to trigger increased LSK cell proliferation in order to maintain absolute cell numbers and to generate more LK cells that in turn eventually leads to HSC exhaustion.

To assess the contribution of cell death to the embryonic lethal phenotype, we overexpressed BCL2 in fetal liver stem cells by intercrossing *Chk1*$^{fl/-}$ *Vav-Cre* with *Vav-BCL2* transgenic mice or by co-deleting *p53*. Consistent with a failure of *p53* loss to restore development of *Chk1*-deficient embryos [21,22], we also failed to detect restoration of hematopoiesis in *Chk1*$^{fl/-}$ *Vav-Cre p53*$^{-/-}$ embryos (Fig EV4A). Moreover, we also failed to observe a rescue of hematopoiesis upon BCL2 overexpression (Figs 7A and EV4A), indicating that even if cell death was blocked, these cells are unable to expand. Loss of p53 or overexpression of BCL2 had also no impact on the changes in cKit expression observed upon loss of *Chk1* (Figs 7A and EV4B).

As seen before, we again noted a drop in LSK CD34⁻Flt3⁻ LT-HSCs combined with an increase in the percentage of LSK CD34⁺Flt3⁻ ST-HSCs and CD34⁺Flt3⁺ MPPs. BCL2 overexpression did not change the percentage of these cells (Fig 7B). Despite an observable trend toward an increase in surviving BCL2 transgenic LSK^hi and LSK^lo as well as LK^lo cells (Fig EV4B), fetal liver cells

cryosections from fetal livers by TUNEL-staining or FACS-sorted LSK cells to assess signs of DNA damage by intracellular staining for γH2A.X. Consistent with our findings *in vitro*, we found a strong

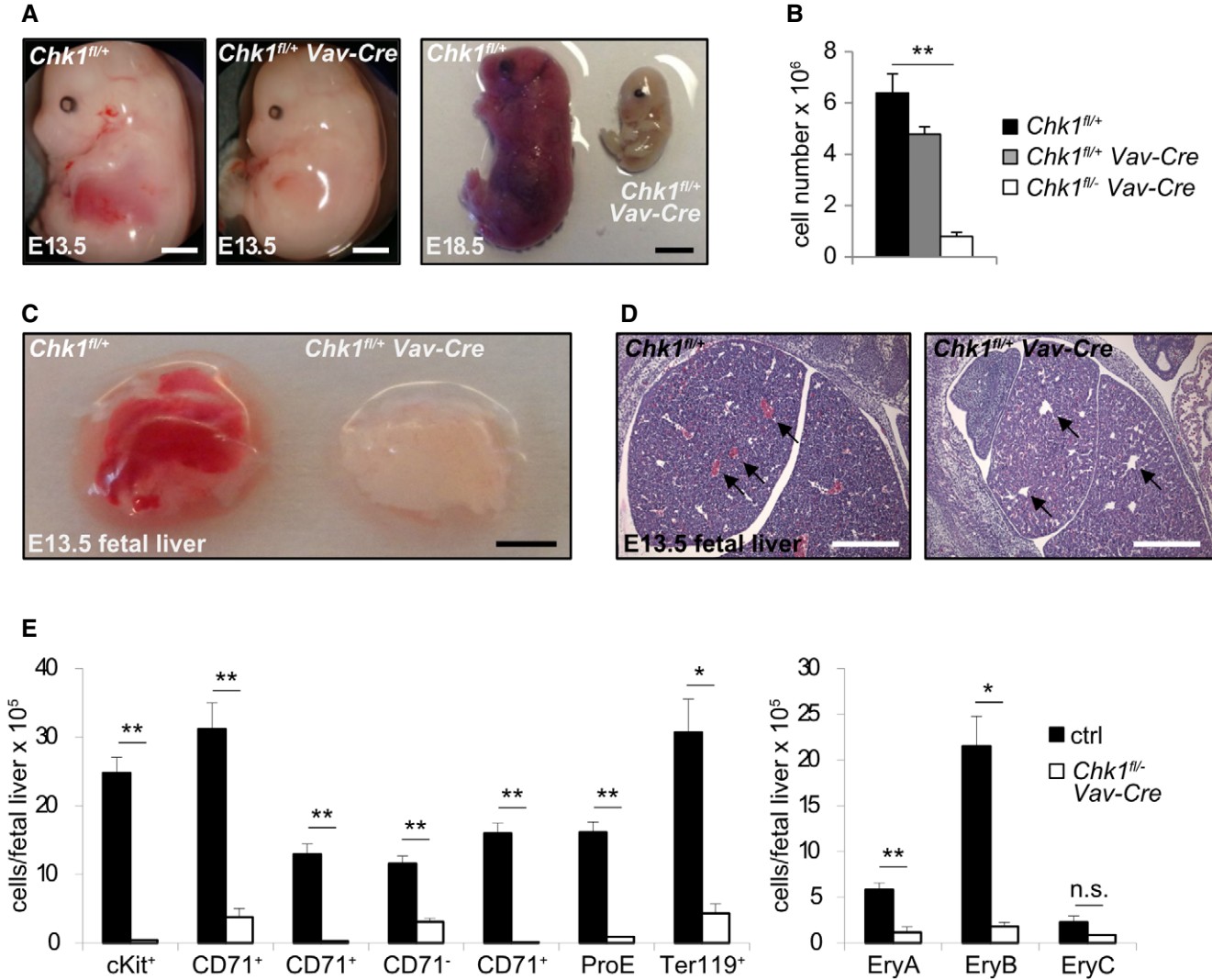

**Figure 4. Deletion of *Chk1* in HSC prevents fetal hematopoiesis.**

A   Representative pictures of E13.5 and E18.5 embryos of the indicated genotypes. Scale bar E13.5 = 1 mm, E18.5 = 5 mm.

B   Quantification of fetal liver cell number at E13.5. Bars represent means ± s.e.m. (N = 4 for *Chk1*<sup>fl/+</sup>, N = 3 for *Chk1*<sup>fl/+</sup> *Vav-Cre*, and N = 4 for *Chk1*<sup>fl/−</sup> *Vav-Cre*).

C   Representative pictures of isolated fetal livers at E13.5. Scale bar = 1 mm.

D   Representative H&E staining of fetal liver sections from E13.5 embryos. Black arrows indicate lack of erythroid cells in the blood vessels of *Chk1*<sup>fl/−</sup> *Vav-Cre* fetal livers. Scale bar = 500 μm.

E   Single cell suspensions of fetal livers from *Chk1*<sup>fl/+</sup>, *Chk1*<sup>fl/+</sup> *Vav-Cre*, and *Chk1*<sup>fl/−</sup> *Vav-Cre* embryos were stained with antibodies recognizing cKit, CD71, Ter119, or CD45 to assess primitive and definitive erythropoiesis by flow cytometry. Large "EryA" erythroblasts (CD71<sup>high</sup> Ter119<sup>high</sup> FSC<sup>high</sup>), smaller, more mature "EryB" erythroblasts (CD71<sup>high</sup> Ter119<sup>high</sup> FSC<sup>low</sup>); most mature erythroblast subset is EryC (CD71<sup>low</sup> Ter119<sup>high</sup> FSC<sup>low</sup>). Bars represent means ± s.e.m. ctrl N = 5 (pooled N = 4 *Chk1*<sup>fl/+</sup> and N = 1 *Chk1*<sup>fl/+</sup> *Vav-Cre*), N = 3 for *Chk1*<sup>fl/−</sup> *Vav-Cre*.

Data information: Asterisks indicate significant differences: *P < 0.05, **P < 0.01 using unpaired Student's t-test.

overexpressing BCL2 were again unable to give rise to colonies in methocult assays (Fig 7C). This suggests that HSPCs that may survive the loss of CHK1 in the presence of high BCL2 levels fail to expand and undergo cell cycle arrest, as these cells accumulate even more DNA damage when compared to *Chk1*<sup>fl/−</sup> *Vav-Cre* cells (compare Figs 6C and EV4C). Consistently, Western analysis performed on E14.5 fetal liver cells confirmed that expression levels of CHK1 were drastically reduced in knock-out embryos, as was the expression of Cyclin D (Fig 7D), suggesting that the residual fetal liver

cells present were not cycling. Notably, BCL2 overexpression did not affect the loss of Cyclin D, confirming that when cell death is blocked, these cells still fail to thrive.

Taken together, our findings suggest that HSPCs depend on CHK1 to prevent the accumulation of DNA damage during cell cycle progression during fetal liver colonization and that in the absence of CHK1 these cells undergo apoptosis. Yet, when intrinsic apoptosis is impaired, cell cycle arrest prevents HSPC expansion as a second barrier to prevent subsequent mutational spread.

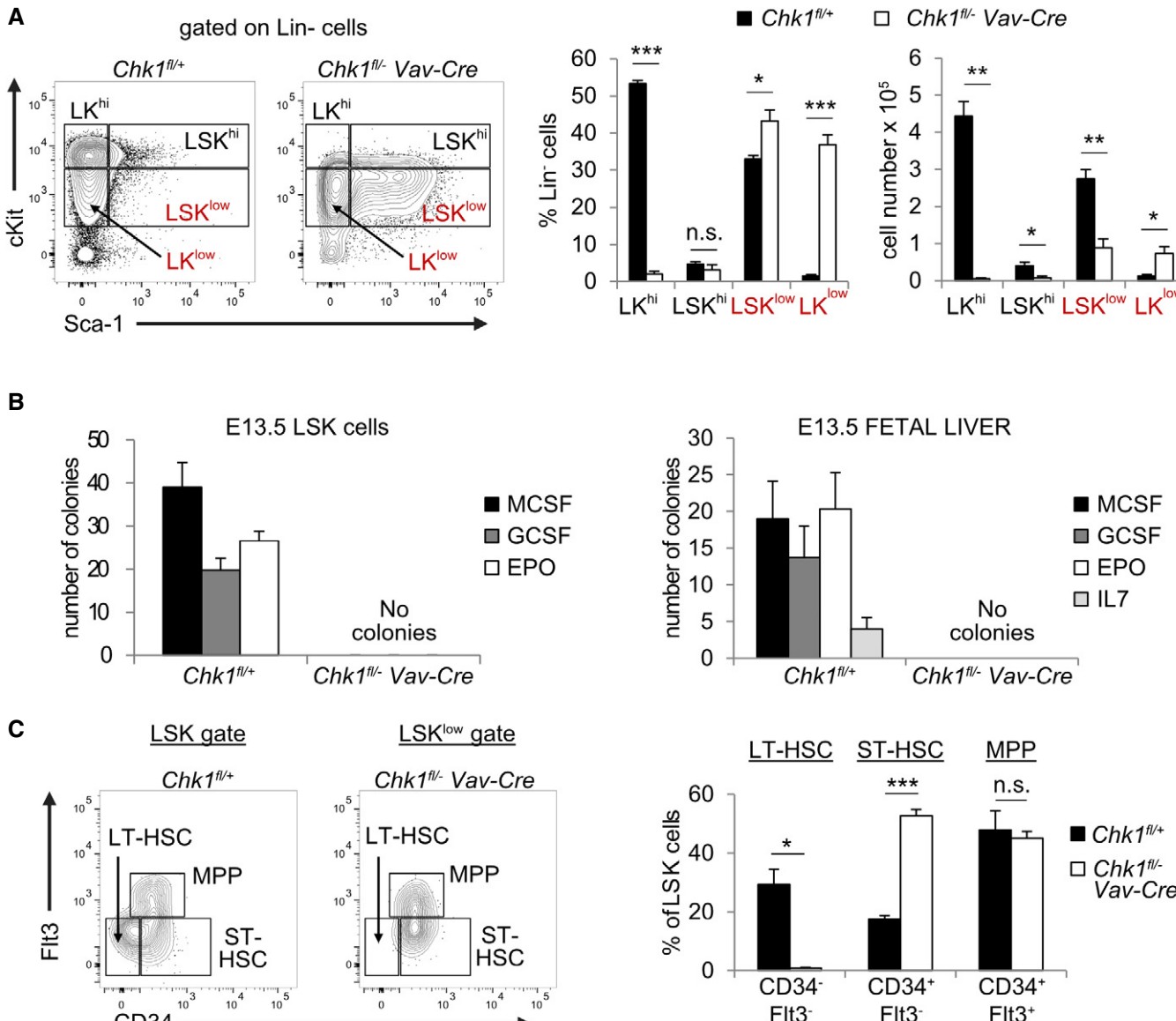

**Figure 5. CHK1 deficiency causes stem cell loss in the fetal liver.**

A   Representative dot-plots of the LSK cell phenotype of *Chk1*$^{fl/+}$ and *Chk1*$^{fl/-}$ *Vav-Cre* embryos observed in E13.5 fetal livers and relative distribution/absolute numbers of LK$^{hi}$ and LSK$^{hi}$ cells, as well as LK$^{low}$ and LSK$^{low}$ cells, found in the fetal liver on E13.5. Bars represent means ± s.e.m. $N = 4$ for *Chk1*$^{fl/+}$ and $N = 4$ for *Chk1*$^{fl/-}$ *Vav-Cre*.

B   Colony formation potential of FACS-sorted LSK cells (left panel, $N = 4$ for *Chk1*$^{fl/+}$ and $N = 5$ for *Chk1*$^{fl/-}$ *Vav-Cre*) and total fetal liver cells (right panel, $N = 3$ for *Chk1*$^{fl/+}$ and $N = 6$ for *Chk1*$^{fl/-}$ *Vav-Cre*) in methyl cellulose assays. Bars represent means ± s.e.m.

C   Flow cytometric analysis of LSK$^{hi}$ or LSK$^{low}$ cells stained with antibodies specific for CD34 and Flt3 ($N = 4$ for *Chk1*$^{fl/+}$ and $N = 4$ for *Chk1*$^{fl/-}$ *Vav-Cre*), to discriminate LT- and ST-HSC from MPP within the LSK subset. Bars represent means ± s.e.m.

Data information: Asterisks indicate significant differences: *$P < 0.05$, **$P < 0.01$, ***$P < 0.001$ using unpaired Student's *t*-test.

## CHK1 loss is selected against upon conditional deletion in adult mice

To establish the role of CHK1 in adult hematopoiesis, we aimed to conditionally delete *Chk1* in adult mice by taking advantage of tamoxifen (TAM)-inducible and hematopoiesis-restricted *Cre* transgene expression in *Vav-Cre*$^{ERT2}$ mice [39]. To monitor recombination in hematopoietic cells by flow cytometry, we additionally introduced a CRE-reporter allele (mT/mG) in the *ROSA26* locus that enables expression of membrane-targeted Tomato prior to and membrane-targeted GFP after CRE-mediated excision of a *loxP* flanked STOP cassette [40]. From these animals (mT/mG ± *Chk1*$^{fl/fl}$), Hoxb8-FL cells were established and transduced with a retrovirus expressing CRE. GFP expression levels were followed over time, and GFP$^+$ and Tomato$^+$ cells were sorted on day 2 after transduction. Under these conditions, DAPI-negative viable cells expressing GFP

were found to lack CHK1 protein, but showed clear signs of DNA damage (Fig 8A). Similar findings were made when Hoxb8-FL cells from *Chk1*^fl/fl^ mice were transduced with a retrovirus harboring a

CRE-IRES-GFP cassette and GFP⁺ cells were sorted for Western blotting on day 3 (Fig 8B). Of note, GFP⁺ cells were rapidly lost over time (Fig 8A), suggesting rapid cell death in the absence of CHK1.

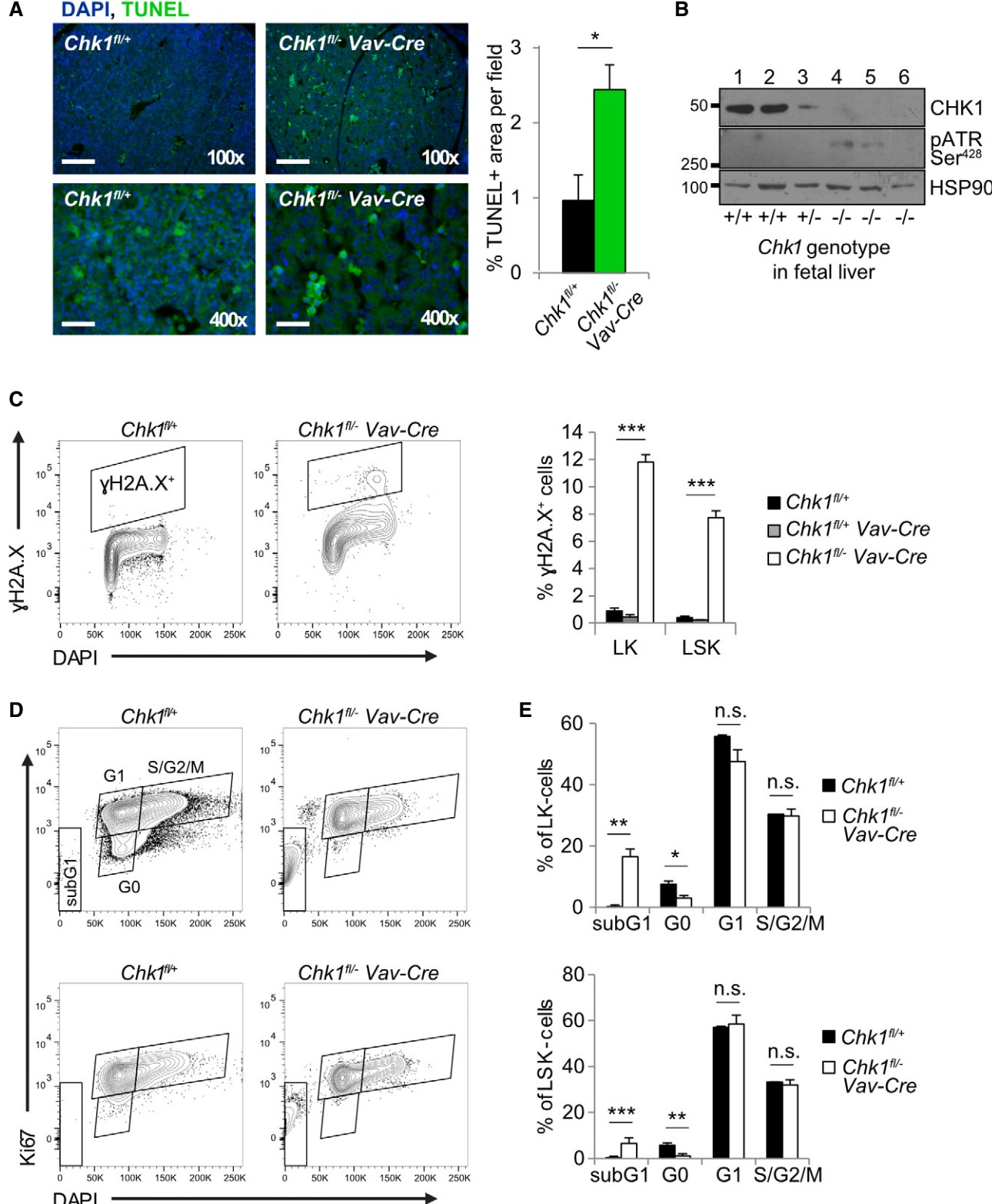

**Figure 6.**

Figure 6. CHK1-deficient HSPCs accumulate DNA damage during compensatory proliferation triggering cell death.

A   Terminal deoxynucleotidyl transferase dUTP nick end labeling (TUNEL) staining of fetal liver cryosections of the indicated genotypes at E13.5 and quantification. Scale bar 100× = 200 μm, 400× = 50 μm. The percentage of TUNEL+ area/field was assessed across three random fields from two control ($Chk1^{fl/+}$) and two knock-out embryos ($Chk1^{fl/-}$ *Vav-Cre*) using *ImageJ*, as reported by us before [60]. Bars represent means ± s.d.

B   Total fetal liver cell suspensions of $Chk1^{fl/+}$ (+/+), $Chk1^{fl/+}$ *Vav-Cre* (+/−), and $Chk1^{fl/-}$ *Vav-Cre* (−/−) E13.5 embryos were processed for Western blot analysis using the indicated antibodies. Each lane represents samples isolated from an individual littermate embryo.

C   DAPI-negative viable LK and LSK cells of the indicated genotypes (littermates) were sorted from E13.5 fetal livers and immediately fixed in 70% EtOH. Fixed cell suspensions were stained for γH2A.X and DAPI intracellularly. Shown here are representative dot-plots and quantification of $N$ = 4 embryos per genotype; Bars represent means ± s.e.m.

D, E   Total E13.5 fetal liver cells were processed for intracellular Ki67 staining in combination with cell surface antibody staining to discriminate LK (top panel) from LSK (lower panel) cells. Shown here are representative dot-plots of Ki67 and DAPI staining in LK and LSK cells, quantified in (E). Bars represent means ± s.e.m. $N$ = 4 for $Chk1^{fl/+}$ and $N$ = 5 for $Chk1^{fl/-}$ *Vav-Cre*.

Data information: Asterisks indicate significant differences: *$P$ < 0.05, **$P$ < 0.01, ***$P$ < 0.001 using unpaired Student's *t*-test.
Source data are available online for this figure.

Yet, Western analysis is of limited sensitivity and PCR analysis amplifying the conditional allele still scored positive in FACS-sorted GFP$^+$ cells, indicating the presence of cells that had retained at least one functional *Chk1* allele (Appendix Fig S2).

Next, compound mutant mice were treated with tamoxifen by oral gavage for 5 days and reporter activity was monitored in the blood 2 or 9 days later (day 7 and day 14). In general, tamoxifen treatment led to a reduction in total white blood cell (WBC) counts, affecting lymphoid and myeloid cells alike, but this was not related to the genotype. In addition, mice carrying the conditional *Chk1* allele presented with a transient drop in platelet number on day 7 that was no longer visible on day 14 of analysis (Fig EV5).

Tracing CRE-mediated recombination by PCR and GFP expression in primary and secondary hematopoietic organs revealed that up to 25% of myeloid cells in the peripheral blood were GFP$^+$ but only about 10% of T or B cells showed reporter expression (Fig 8C and D, Table EV1, Appendix Fig S3). In the bone marrow, the percentage of LSK and LK cells was comparable between genotypes (Fig 8E) and an average of 40% of LSK and LK cells was found GFP$^+$ indicating deletion of *Chk1* by day 15 (Fig 8F; Table EV1). Yet, with the exception of a transient reduction of GFP$^+$ LK cells on day 8 in $Chk1^{fl/fl}$ *Vav-Cre*$^{ERT2}$ mice (Fig 8F), suggesting increased vulnerability, no significant difference was seen between the genotypes. Within the GFP-positive fraction, the distribution of Lin$^-$ cells as well as that of LK or LSK cells was again comparable between genotypes, suggesting that timed deletion of CHK1 might be well-tolerated in bone marrow-resident LSK and LK cells of adult mice (Table EV1).

Of note, the percentage of GFP$^+$ cells gradually decreased with maturation stage, but this was seen in both genotypes tested, demonstrating only the gradual and time-dependent turn-over of the hematopoietic system (Table EV1). Consistently, in the bone marrow, up to 37% of IgM$^-$AA4.1$^+$ pro/pre-B and 27% of IgM$^+$AA4.1$^+$ naïve B cells, but only 2% of recirculating IgM$^+$D$^+$ B cells were found to be GFP$^+$ on day 15. Accordingly, only about 13% of splenic IgM$^+$D$^-$ B cells were GFP$^+$, indicating CRE deletion, while up to 30% myeloid cells in the spleen or bone marrow switched from Tomato to GFP expression (Table EV1). The percentage of GFP-positive thymocytes carrying the conditional $Chk1^{fl/fl}$ allele was ranging between 12 and 22% on day 15 but only about 3% of all mature CD4$^+$ or CD8$^+$ T cells expressed GFP in the periphery (Table EV1, Appendix Fig S3).

Remarkably, however, within the GFP$^+$ subset that has activated CRE recombinase, the distribution of T cells, B cells, and myeloid cells was always comparable between genotypes, suggesting that loss of CHK1 did not perturb leukocyte homeostasis *in vivo*, at least not within our observation period of 15 days. Alternatively, GFP$^+$ cells may not have deleted both *Chk1* alleles, despite recombining the reporter allele. To monitor deletion at the protein level, GFP$^+$ and Tomato$^+$ fractions were sorted from the bone marrow on day 8 and the thymus on day 15 after the first oral gavage. Bone marrow samples from three animals were pooled to obtain cell numbers suitable for Western analysis (Fig 8G). Remarkably, while the expression of CHK1 was significantly reduced in GFP$^+$ bone marrow cells on day 8, GFP$^+$ and Tomato$^+$ thymocytes showed highly comparable CHK1 protein levels on day 15 after the first administration of tamoxifen (Fig 8H). This suggests that only cells that manage to maintain CHK1 expression are able to reach the thymus, and together, these findings indicate that CHK1 expression is essential for the maintenance of a functional hematopoietic system in the adult.

Figure 7. Apoptosis inhibition cannot restore hematopoiesis in the absence of CHK1.

A   Representative dot-plots and quantification of E13.5 fetal liver LSK$^{hi}$/LSK$^{low}$ cell phenotypes observed in embryos of the indicated genotypes. $N$ = 21 for $Chk1^{fl/+}$, $N$ = 5 for $Chk1^{fl/-}$ *Vav-Cre*, $N$ = 16 for $Chk1^{fl/+}$ *Vav-BCL2*, $N$ = 5 for $Chk1^{fl/-}$ *Vav-Cre Vav-BCL2*, $N$ = 2 for $p53^{-/-}$ $Chk1^{fl/+}$, and $N$ = 4 for $p53^{-/-}$ $Chk1^{fl/-}$ *Vav-Cre*.

B   Flow cytometry-based analysis of LSK cells stained with antibodies specific for CD34 and Flt3, to discriminate LT- and ST-HSC from MPP within the LSK subset in fetal livers of embryos of the indicated genotypes. Left: $N$ = 25 for $Chk1^{fl/+}$, $N$ = 7 for $Chk1^{fl/+}$ *Vav-Cre*, and $N$ = 7 for $Chk1^{fl/-}$ *Vav-Cre*. Right: $N$ = 16 for $Chk1^{fl/+}$ *Vav-BCL2*, $N$ = 6 for $Chk1^{fl/+}$ *Vav-Cre Vav-BCL2*, and $N$ = 5 for $Chk1^{fl/-}$ *Vav-Cre Vav-BCL2*.

C   Analysis of the colony formation potential of E14.5 total fetal liver cells of the indicated genotypes in methylcellulose. $N$ = 3 for controls (pooled; 1 $Chk1^{fl/+}$, 2 $Chk1^{fl/+}$ *Vav-Cre*), $N$ = 3 for $Chk1^{fl/-}$ *Vav-Cre*, and $N$ = 3 for $Chk1^{fl/-}$ *Vav-Cre Vav-BCL2*.

D   Western blot analysis of fetal liver cells of the indicated genotypes (each lane represents an embryo) at E14.5 using the indicated antibodies.

Data information: Bars usually represent means ± s.e.m. Bars depicting $p53^{-/-}$ $Chk1^{fl/+}$ and $p53^{-/-}$ $Chk1^{fl/+}$ *Vav-Cre* show means ± s.d. due to the lower number of mice analyzed. Asterisks indicate significant differences: *$P$ < 0.05, **$P$ < 0.01, ***$P$ < 0.001 using unpaired Student's *t*-test.
Source data are available online for this figure.

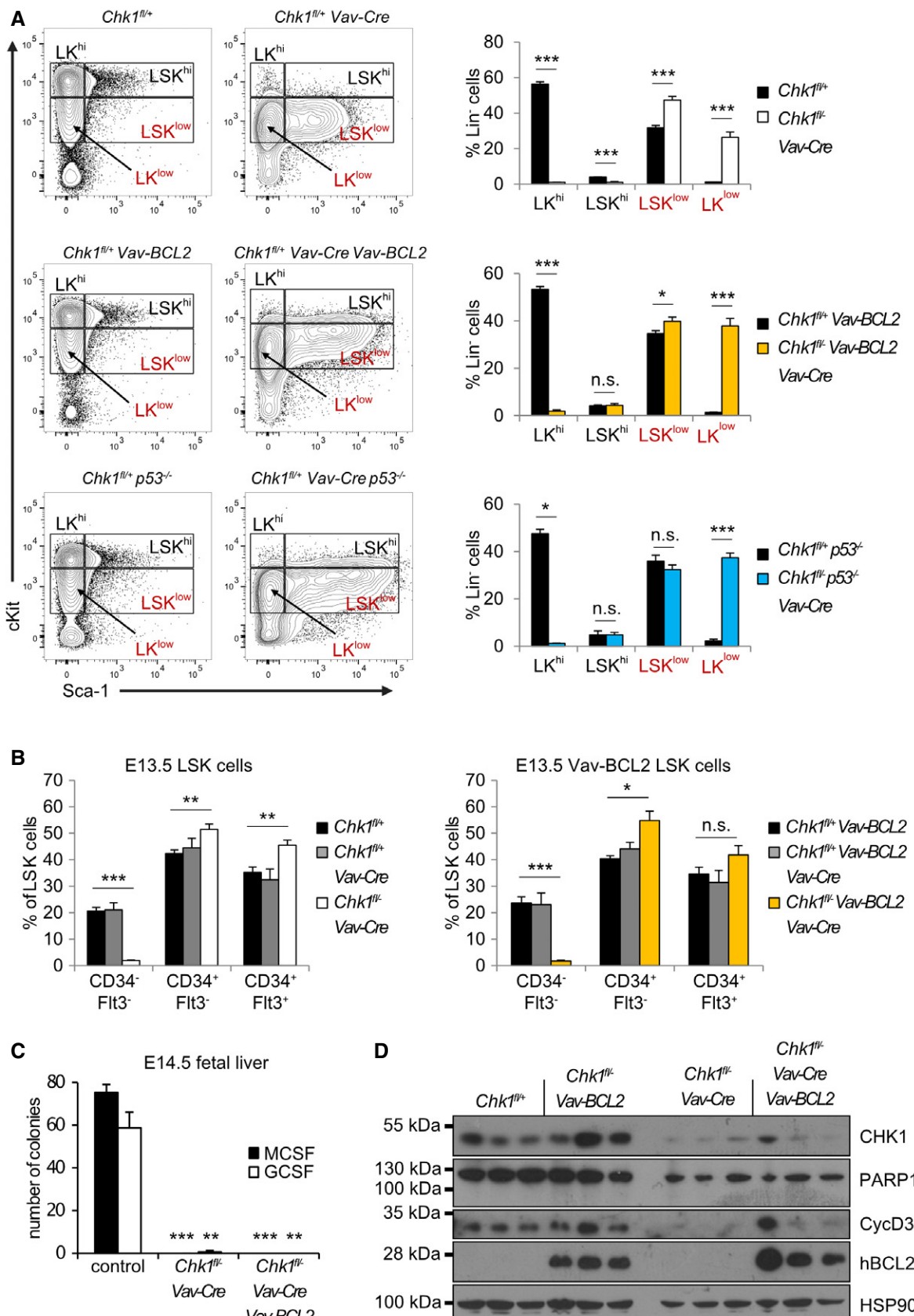

Figure 7.

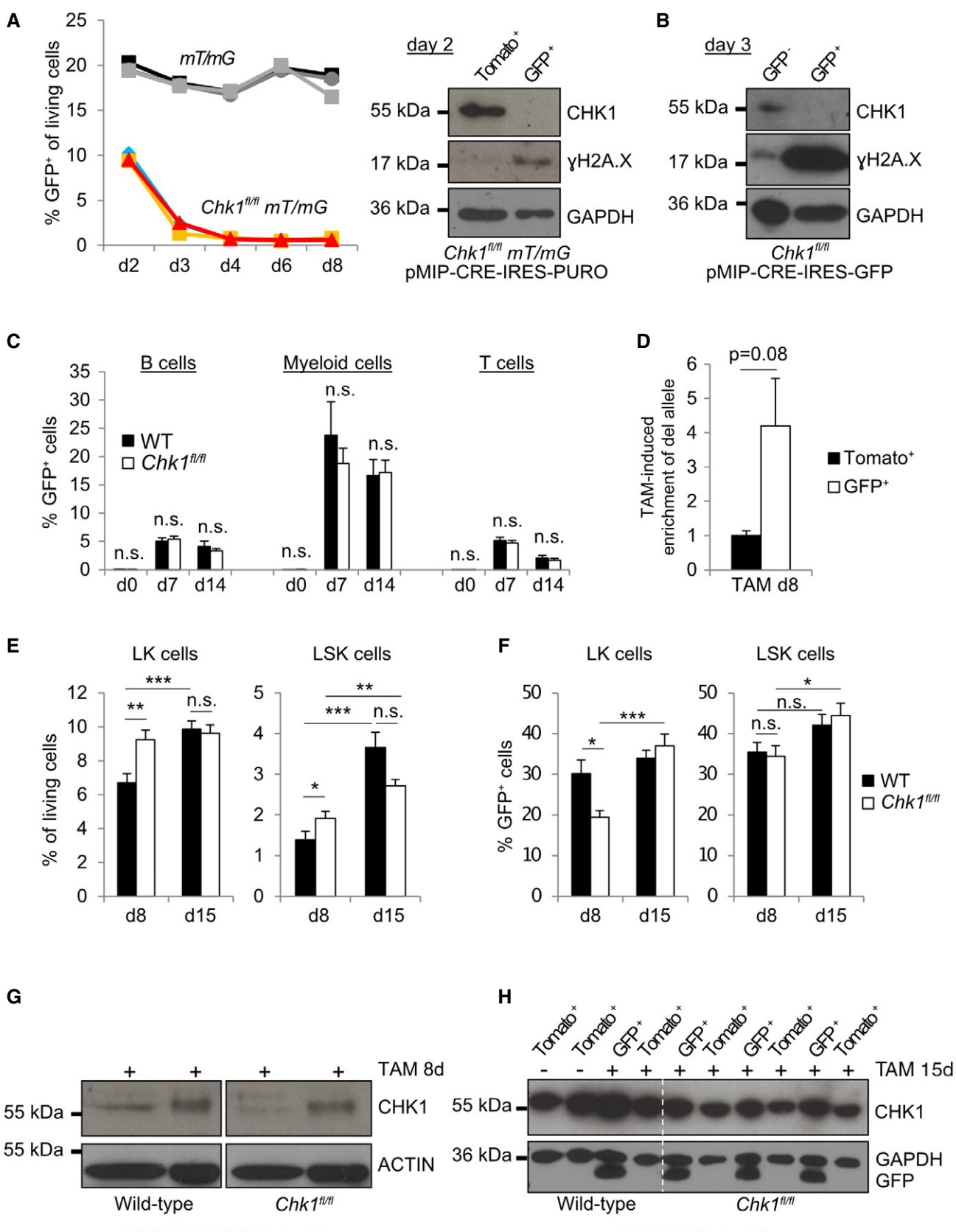

Figure 8.

**Figure 8.** **Acute depletion of *Chk1* in the hematopoietic system leads to cell loss *in vitro* and retention of CHK1 expression *in vivo*.**

A   Three independent *Chk1*$^{fl/fl}$ *mT/mG* (colored) and *mT/mG* (gray) Hoxb8-FL progenitor cell lines were transduced with a retroviral construct carrying Cre (pMIP-CRE-IRES-PURO) and monitored over time for GFP expression. Shown here are three technical replicates per genotype. Additionally, *Chk1*$^{fl/fl}$ *mT/mG* Hoxb8-FL progenitor cells were FACS-sorted for GFP$^+$ and dsRED$^+$ cells 48 h after transduction with the Cre-construct. Sorted cells were processed for Western blotting using the indicated antibodies.

B   *Chk1*$^{fl/fl}$ Hoxb8-FL progenitor cells were transduced with a retroviral construct carrying Cre and GFP (pMIG-CRE-IRES-GFP). Cells were FACS-sorted 72 h after transduction with the retroviral construct and processed for Western blotting using the indicated antibodies.

C   Percentage of GFP$^+$ cells indicative for CRE expression and recombination in the peripheral blood after tamoxifen administration by gavage (5 × 2 mg, 200 μl/mouse; daily). Bars represent means ± s.e.m. (Day 0 *N* = 9/10, WT/*Chk1*$^{fl/fl}$, Day 7 *N* = 8/7, Day 14 *N* = 3/3).

D   Quantification of *Chk1*-del PCR-product enrichment in DNA from GFP$^+$ vs. Tomato$^+$ bone marrow cells sorted on day 8 after first TAM administration. Transgenic Cre levels were quantified in parallel for reference. Bars represent means ± s.e.m. from *n* = 5 *mT/mG* Vav-Cre$^{ERT2}$ *Chk1*$^{fl/fl}$ mice.

E, F   (E) Percentage of living LK or LSK cells and (F) percentage of GFP$^+$ LK or LSK cells found in the bone marrow of *mT/mG* Vav-Cre$^{ERT2}$ (WT) and *mT/mG* Vav-Cre$^{ERT2}$ *Chk1*$^{fl/fl}$ (*Chk1*$^{fl/fl}$) mice on day 8 or 15 after first TAM administration. Bars represent means ± s.e.m. (Day 8 *N* = 8/8, WT/*Chk1*$^{fl/fl}$, Day 15 *N* = 7/8).

G   GFP$^+$ (CRE) and Tomato$^+$ (CRE$^-$) total bone marrow cells were sorted on day 8 after first TAM administration and processed for Western blot using the indicated antibodies. Sorted cells from three animals—*mT/mG* Vav-Cre$^{ERT2}$ (WT) and *mT/mG* Vav-Cre$^{ERT2}$ *Chk1*$^{fl/fl}$ (*Chk1*$^{fl/fl}$)—were pooled per lane.

H   Thymocytes of the indicated genotypes were sorted based on GFP$^+$ and Tomato$^+$ on day 15 after the first TAM administration for Western analysis using the indicated antibodies.

Data information: Asterisks indicate significant differences: *$P$ < 0.05, **$P$ < 0.01, ***$P$ < 0.001 using unpaired Student's *t*-test.
Source data are available online for this figure.

## Discussion

HSC are responsible for the supply of blood cells throughout our lifetime. Therefore, it is essential to control their stemness in response to proliferative cues to avoid stem cell exhaustion driving hematopoietic failure and to prevent accumulation of DNA damage or mutations that might contribute to malignant disease with age. CHK1 is a key regulator of DNA-replication fidelity and the G2/M transition upon DNA damage, critical in early embryogenesis and meanwhile a recognized target in cancer therapy [11,41]. Yet, the impact of CHK1 loss or inhibition on normal hematopoiesis remains largely unknown.

We could recently show that human Burkitt's lymphoma and pre-B ALL cell lines, similar to primary pre-B and mature B cells from mice, die an apoptotic cell death controlled by the BCL2 family when treated with CHK1 inhibitors [24]. Similarly, as reported here, MPP-like Hoxb8-FL cells undergo mitochondrial apoptosis when treated with CHK1i. This death is clearly BAX/BAK dependent and can be delayed by BCL2 overexpression (Figs 1 and 2). Remarkably, despite the induction of DNA damage and subsequent p53 stabilization, these cells are not protected from death by loss of p53. This suggests that NOXA and PUMA, two recognized pro-apoptotic p53 target genes that act upstream of BAX/BAK [42], may only play a limited role in this apoptotic response. Instead, double depletion of Bim and Noxa (or to a minor extent Bim and Puma) could delay cell death similar to BCL2-overexpression *in vitro* (Fig EV1); however, the precise mechanism of BAX/BAK activation in response to CHK1 inhibition remains to be investigated in detail. Of note, the mode of cell death seems to be conserved in human CD34$^+$ HSPCs (Fig 3).

Our *in vivo* results clearly demonstrate that CHK1 is an essential component of the regulatory network controlling HSPC development and expansion (Fig 4). Conditional deletion of CHK1 in definitive HSC (dHSC) using Vav-CRE-mediated recombination causes failure of fetal hematopoiesis, associated with a reduction in fetal liver cell number, loss of cKit expression, and impaired hematopoiesis, detectable as early as on embryonic day 12.5 of gestation in *Chk1*$^{fl/-}$ Vav-Cre embryos (FS & AV; personal observations). Fetal livers of these embryos present with a relative

accumulation of ST-HSC and MPPs at the cost of LT-HSC (Fig 5) that are possibly driven into compensatory proliferation cycles and hence accumulate high levels of DNA damage (Fig 6). This DNA damage may trigger cell death of fetal liver-resident LSK cells that are cKit$^{low}$ accounting for the embryonic lethality seen in *Chk1*$^{fl/-}$ Vav-Cre mice.

In support of this notion, we could detect only ~ 6% cKit$^+$ cells in *Chk1*$^{fl/-}$ Vav-Cre fetal livers at E13.5 compared to more than 40% in littermate controls (Fig EV4). Although cKit cannot be used as a general marker for dHSC and their progeny, as it is present on a variety of other cell types, it can be used to discriminate primitive HSC (pHSC) as well as primitive erythrocytes from dHSC and definitive erythrocytes, as the former lack cKit, as reviewed in Ref. [2]. Furthermore, it was shown that only cKit$^+$ cells, but not cKit$^{low}$ or cKit$^-$ cells from the E11.5 AGM region or the fetal liver, can reconstitute lethally irradiated mice confirming that definitive hematopoiesis relies on the SCF receptor, cKit [43]. Together, this suggests that loss of CHK1 using Vav-Cre, activated as early as on day E11.5 [44], results in the loss of definitive hematopoiesis in the fetal liver and that the blood cells we can detect in *Chk1*$^{fl/-}$ Vav-Cre embryos are of primitive hematopoietic ancestry.

Interestingly, LSK cells isolated from the fetal liver did not massively die in culture when treated with the CHK1 inhibitor while total fetal liver cells did die in a dose-, time-, and BCL2-dependent manner (Fig 2), suggesting that loss of CHK1 dominantly affects the survival of actively cycling HSPCs. This may be explained by the fact that all these cells have a higher proliferative index compared to LSK cells. An alternative explanation would be the lack of CHK1 expression in fetal liver-resident LSK cells, rendering them insensitive to CHK1 inhibitors. Indeed, mRNA expression analysis showed lowest levels of *Chk1* mRNA in resting bone marrow LSK cells but higher levels in fetal liver LSKs and mobilized LSKs or MPPs [29], as well as increased CHK1 protein in expanding human CD34$^+$ HSPCs [35]. Moreover, mobilized LSK cells showed clear signs of replication stress-associated CHK1 activation and hallmarks of DNA damage [30], indicating the need for CHK1 to deal with replication stress in cycling HSPCs that would otherwise trigger apoptosis. This notion is further supported by the fact that we observed a 7-fold decrease

in total fetal liver cell number (Fig 4B) in the absence of CHK1 but normal absolute numbers of LSK cells (Fig 5A). It seems that in the absence of CHK1-mediated cell cycle control, LT-HSC lose self-renewal potential and exhaust (Fig 5) because they are recruited into the pool of transient amplifying ST-HSC and MPP. These cells then arrest their cell cycle in the presence of accumulating DNA damage when unable to die. Indeed, we could observe the induction of cell cycle arrest and *p21* mRNA when we tested BCL2-overexpressing or BAX/BAK-deficient Hoxb8-FL cells in response to CHK1i that cannot undergo apoptosis (Fig 1). Together, this might explain why overexpression of BCL2 in fetal liver HSPCs fails to rescue hematopoiesis in the absence of CHK1 (Fig 7), despite protecting HSPCs from cell death *in vitro* (Fig 2).

Similar to what we have noted in MPP-like cells and fetal liver cell cultures, deletion of CHK1 in the small intestine, using CYP1A1 promoter to drive CRE expression in *AhCre* mice, leads to p53-independent apoptosis due to massive DNA damage. Remarkably, the GI tract can be restored in these mice from stem cells that fail to delete *Chk1* [45], a phenomenon not seen in *Chk1$^{fl/-}$ Vav-Cre* embryos but clearly visible in animals where we attempted conditional deletion of CHK1 (Fig 8). In this case, reporter activity failed to faithfully read out target gene deletion, as noted by others in various settings [46], suggesting essential roles for CHK1 also in adult hematopoiesis. In line with these observations, developing mammary epithelial cells and early thymocytes do undergo cell death when CHK1 is deleted using *WAP-Cre* or *Lck-Cre*, respectively [25,47]. Furthermore, deletion of CHK1 in B-cell progenitors using *Mb1-Cre* blocks B-cell development at the pro-B-cell stage in the bone marrow [24]. Of note, neither B-cell nor T-cell development could be restored by BCL2 overexpression, indicating concomitant induction of cell cycle arrest. This response was documented in cycling pre-B cells unable to undergo apoptosis upon CHK1i treatment [24]. Collectively, this suggests that deletion of CHK1 is lethal for cycling cell types, with the notable exception of chicken DT40 B lymphoma cells [48]. Consistently, adult hepatocytes are unaffected by loss of CHK1, most likely as homeostatic hepatocyte proliferation is minimal [49,50].

Of note, depletion of other components of the cell cycle network controlling faithful chromosome segregation and safe passage through M-phase give phenotypes similar to those caused by CHK1 deletion. It was shown that loss of a single allele of the spindle assembly checkpoint component, *Mad2l1*, results in growth deficits of immature hematopoietic progenitor cells [51]. Furthermore, deletion of *survivin*, a member of the chromosomal passenger complex (CPC) in adult hematopoietic cells, results in bone marrow aplasia and mortality of hematopoietic progenitor cells whereas *survivin$^{+/-}$* mice show problems in erythropoiesis [52]. In line with that, it was shown that Survivin-deficient hematopoietic progenitor cells fail to form colonies of the erythroid and megakaryocyte lineage [53]. Moreover, conditional Aurora A kinase deletion using *Mx1-Cre* in a bone marrow reconstitution approach caused a severe cell-autonomous death in nearly all hematopoietic lineages [54]. Together, this suggests that manipulation of the cell cycle machinery allowing for premature entry in or exit from mitosis results in hematopoietic failure or triggers a severe growth disadvantage in hematopoietic cells.

As inhibitors of CHK1 and related cell cycle regulating proteins such as Aurora kinase A or MAD2L1 are currently under investigation in clinical trials to treat various malignancies (as reviewed in Ref. [41]), it is important to understand whether their transient

inhibition can be tolerated temporarily or causes myeloablation, as seen frequently in response to chemotherapy. Studies where MAD2L1 is deleted in adult mice are still missing but would be of tremendous value to address their suitability as drug targets. *Chk1$^{+/-}$* animals on C57BL/6N background, however, were phenotypically normal and also did not show signs of anemia within their first year of life (F.S. & A.V.; personal observations), opening an opportunity to target cancer cells that appear to be even more dependent on CHK1 function for survival [20,55]. This idea is supported by a recent study testing a new CHK1i for its efficacy to kill AML in mice that proved particularly effective when combined with AraC and G-CSF [56]. Here, the authors noted no negative side effects on the HSC or MPP compartment, suggesting that transient inhibition, in contrast to chronic (genetic) deletion, might indeed be tolerated, and therefore opening a window of opportunity for drug treatment. Nonetheless, our findings raise the possibility that CHK1i treatment may become problematic during pregnancy as well as other situations that call stem cells out of dormancy, such as bacterial or viral infection, as well as chemotherapy-based combination regimens. Last but not least, CHK1i-based treatments could also contribute to secondary malignancies triggered by DNA damage in treatment-surviving cells.

## Materials and Methods

### Mice

Animal experiments were performed in accordance with Austrian legislation (BMWF: 66-011/0106-WF/3b/2015) and in agreement with the animal ethics committee of the Medical University of Innsbruck. The generation and genotyping of *Chk1$^{fl/fl}$, p53$^{-/-}$ Vav-BCL2, Vav-iCre,* and *Vav-CRE-ERT2* mice have been described [36,39,48,57,58,59]. All mice were maintained or backcrossed on a C57BL/6N genetic background. Mice were housed in ventilated cages with nesting material and were maintained on a 12:12-h light:-dark cycle. If not stated differently, mice were analyzed at the age of 8–16 weeks. 200 mg tamoxifen was dissolved in 500 μl 100% EtOH and subsequently mixed with 19.5 ml sterile filtered sunflower oil while intensive vortexing. 1 ml aliquots were frozen at −20°C. Prior use, aliquots were thawed at 37°C. Each mouse received 10 mg tamoxifen via gavage (5 × 2 mg/200 μl/day). Cohort A: 12 mice (three experimental mice and three controls analyzed at days 8 and 15 each), Cohort B: 19 mice (five experimental mice and five controls analyzed at day 8; five experimental mice and four controls analyzed at day 8). Semi-blinded cohorts were used meaning it was known that, e.g., three experimental and three control mice were within the group, but the genotype of each individual mouse was not known for the experimenter. Post-mortem mice were re-genotyped via PCR to ensure the correct genotype.

### Cell culture

All murine cells were cultured in RPMI-complete medium: RPMI-1640 medium (Sigma-Aldrich, R0883), supplemented with 10% FCS (Gibco, 10270-106), 2 mM L-glutamine (Sigma, G7513), 100 U/ml penicillin and 100 μg/ml streptomycin (Sigma, P0781), and 50 μM 2-mercaptoethanol (Sigma, M3148). Hoxb8-FL cells were cultured in

RPMI-complete supplemented with 1 μM β-estradiol and 5% supernatant of FLT-3L expressing B16-melanoma cells. Primary murine bone marrow cells, fetal liver cells, or FACS-sorted bone marrow or fetal liver-derived LK or LSK cells were cultured in RPMI-complete media, supplemented with 1 mM sodium pyruvate (Thermo Fisher, 11360070), 1× non-essential amino acids (Thermo Fisher, 11140035), 5% supernatant of FLT-3L expressing B16 melanoma cells, and 2% supernatant of SCF-producing CHO (Chinese hamster ovary) cells. Human cord blood-derived CD34$^+$ HSPCs were cultured in StemPro™-34 SFM 1× serum-free medium (Gibco™; 10639011) supplemented with 10% ES-FCS (Gibco™; 16141061), and human SCF (100 ng/ml, ImmunoTools, 11343325), FLT-3L (100 ng/ml; ImmunoTools 11340035), TPO (50 ng/ml Immuno-Tools, 11344863), and IL-3 (20 ng/ml, ImmunoTools 11340035) cytokines.

### Generation of Hoxb8-FL cells

Fresh bone marrow suspension cells from adult mice were cultivated (1/10$^{th}$ of the cells flushed from 1 tibia and 1 femur) for 2 days in Opti-MEM™ (Gibco, Cat. 31985070) supplemented with 10% FCS, 2 mM L-glutamine, 100 U/ml penicillin, 100 μg/ml streptomycin, 50 μM 2-mercaptoethanol, 10 ng/ml IL-3 (Lot #120948 C2013, PeproTech), 20 ng/ml IL-6 (Lot #090850 A3013, PeproTech), and 2% supernatant of SCF-producing WEHI-231 cells. $2 \times 10^5$ cells were then transduced via 3 × 30 min spin-infection at 37°C (500 $g$, vortex in between) with the Hoxb8-encoding retrovirus using 250 μl Opti-MEM™, 1.25 μl metafectene.

### Reagents

PF-477736 (Selleckchem S2904), CHIR-124 (Selleckchem S2683), QVD (SML0063, Sigma), DMSO (D5879, Sigma), and tamoxifen (Sigma, T5648).

### Flow cytometry and cell sorting

Flow cytometric analysis or cell sorting of single cell suspensions generated from bone marrow or fetal liver was performed on an LSR Fortessa or a FACS-Aria-III, respectively (both BD), and analyzed using FlowJo® v10 software. Antibodies used were as follows: eBioscience lineage-depletion antibodies (B220-bio RA3-6B2, Ter119-bio TER-119, CD3e-bio 145-2C11, CD11b-bio M1/70, Gr-1-bio RB6-8C5), NK1.1-bio (PK136, BioLegend), Sca1-APC (D7, BioLegend), Sca1-PE (D7, BioLegend), cKit-PE/Cy7 (2B8, BioLegend), cKit-FITC (2B8, BioLegend), cKit-PerCP-Cy5.5 (2B8, BioLegend), cKit-BV421 (2B8, BioLegend), CD48-APC (HM48.1, BioLegend), CD150-PE/Cy7 (TC15-12, BioLegend), CD34-eFluor450 (RAM34, eBioscience), Flt3-PE (A2F10, eBioscience), CD45-PE (30-F11; BioLegend), CD71-APC (R17217, eBioscience), B220-FITC (RA3-6B2, BioLegend), B220-APC/eF780 (RA3-6B2, eBioscience), B220-PE (BD RA3-6B2), B220-PerCP-Cy5.5 (RA3-6B2, BioLegend), CD19-BV605 (6D5, BioLegend), IgM-APC (RMM-1, BioLegend), IgM-FITC (BD II/41), IgM-eF450 (eb121-15F9, eBioscience), IgD-PerCP-Cy5.5 (11-26c2a, BioLegend), TCRβ-BV605 (BD, H57-597), TCRβ-FITC (H57-597, eBioscience), CD4-eF450 (GK1.5, eBioscience), CD8-alexa647 (BD, 557682), Mac1-APC (M1/70, eBioscience), CD25-PE (PC61, BioLegend), CD93-PE/

Cy7 (AA4.1, eBioscience), NK1.1-APC (PK136, BioLegend), Ter119-PerCP/Cy5.5 (TER-119) Annexin V-FITC (Lot: B206041, BioLegend), and Annexin V-eF450 (Lot: E11738-1633, eBioscience).

### Nicoletti staining

Cells were fixed in 1 ml 70% ethanol while vortexing and stored at −20°C for a minimum of 60 min. Prior antibody staining, cells were washed twice (800 $g$, 5 min) with 2 ml PBS to remove ethanol. After RNase A (Sigma) digestion (100 mg/ml in PBS, 30 min at 37°C), cells were stained with propidium iodide (40 μg/ml) and the percentages of sub-G1, G1, S, and G2/M cells were monitored by flow cytometry.

### Intracellular staining for γH2A.X

Cells were fixed in ethanol and stored at −20°C. After two washes with PBS, cells were incubated for 15 min in PBS + 0.25% Triton X-100 (Sigma) on ice for permeabilization. Cells were incubated for 60′ with anti-Human/Mouse phospho-H2A.X S139 mAb PerCP-eFluor® 710 (clone CR55T33, eBioscience). Cells were washed in PBS 1%BSA. DAPI (200 ng/ml) was used for DNA content analysis.

### Intracellular Ki67 staining

$10^6$ bone marrow cells were incubated for 30 min at 4°C with the primary antibody mix: lineage-biotin antibodies for B220, CD3, Gr1, Mac1, NK1.1, Ter119, plus antibodies recognizing Sca1-APC or cKit-PE/Cy7 (all 1/100 in PBS + 10%FCS, 50 μl/tube). Cells were washed with 2 ml PBS; then, cells were stained with the secondary antibody mix: streptavidin-PerCP/Cy5.5 (1/100 in PBS + 10%FCS, 50 μl/tube). Cells were washed twice with 2 ml PBS. Cells were fixed and permeabilized using the reagent from the APC-BrdU Flow-Kit (BD, Vienna, Austria), according to the manufacturer's recommendation: Cytofix/Cytoperm, 10′ on ice, 100 μl/tube, Perm/Wash, 1 ml/tube (400 $g$, 5′), Cytofix/Cytoperm-PLUS, 10′ on ice, 100 μl/tube, Perm/Wash, 1 ml/tube (400 $g$, 5′), Cytofix/Cytoperm, 5′ room temperature, Perm/Wash, and 1 ml/tube (400 $g$, 5′). Then, cells were incubated for 30′ on ice with Ki67-alexa488 (16A8, BioLegend), 1/100 in Perm/Wash buffer, and 50 μl/tube. Cells were washed twice with Perm/Wash buffer and 1 ml/tube (400 $g$, 5′), 300 μl DAPI (final concentration 200 ng/ml) diluted in PBS + 10% FCS was added, and cells were analyzed on a flow cytometer.

### Viability assay

Cells were co-stained with Annexin V-FITC, Annexin V-Alexa647, or Annexin V-Pacific blue (BioLegend) 1:1,000 in Annexin V binding buffer (BD) and 7-AAD (1 μg/ml, Sigma). Cell death was analyzed by subsequent flow cytometric analysis.

### Isolation of CD34$^+$ cells

Human cord blood was collected immediately following cesarean births after informed consent of the parents and approval of ethical committee of University Hospital Freiburg, Germany. After Ficoll density gradient-based separation of mononuclear cells, CD34$^+$ cells were isolated with MACS technology (Miltenyi Biotec). The purity

of CD34$^+$ cells was generally greater than 90% as determined via FACS. The purified cells were then frozen in CryoStor CS 10 (Merck, C2874) and stored in liquid nitrogen for later use. After thawing, cells were cultured at a density of $2.5 \times 10^6$ cells/ml overnight StemPro™-34 SFM 1× serum-free medium (Gibco™, #10639011 supplemented with 10% ES-FBS (Gibco™, # 16141061) and human SCF (100 ng/ml, ImmunoTools, 11343325), FLT3-L (100 ng/ml; ImmunoTools, 11343305), TPO (50 ng/ml ImmunoTools, 11344863), and IL-3 (20 ng/ml, ImmunoTools, 11340035). For colony-forming assays, MethoCult™ SF H4436 medium (Stem Cell Technologies) was used. Cells were added with or without CHK1 inhibitors at a density of $10^3$ cells/ml/35 mm cell culture dish and incubated for 10 days. Afterward, colony counts and total cell counts were conducted. Lentiviruses GFP co-expressing pLeGO-iG lentiviral vector, Vsv.g-envelope, and Gag/Pol plasmids were utilized for the production of BCL-2 expressing viruses. As described in Ref. [59], HEK293T cells were used for viral packaging. CD34$^+$ cells were incubated with lentiviruses in the presence of serum-free medium with cytokines for 48 h (MOI = 10 per day).

### Gene expression

RNA was isolated from snap-frozen FACS-sorted or *in vitro* cultivated Hoxb8-FL cells using the Qiagen RNeasy Mini Kit (74104) and the RNase-Free DNase Set (79254) according to the manufacturer's instructions. For each sample, hundred nanograms of RNA was reverse-transcribed to cDNA using the iScript cDNA Synthesis Kit (1708890) according to the manufacturer's instructions. For qRT–PCR with the StepOnePlus System (Applied Biosystems), we used the 2× SYBR Green qPCR Master Mix from BioTool (B21203). Expression levels of *p21* mRNA were normalized to the housekeeping gene *Hprt*. Primers used were as follows (5′–3′): *p21*_Fw AAT TGG AGT CAG GCG CAG AT, *p21*_Rv CAT GAG CGC ATC GCA ATC AC, *Hprt*_Fw GTC ATG CCG ACC CGC AGT C, *Hprt*_Rv GTC CTT CCA TAA TAG TCC ATG AGG AAT AAA C.

### Immunoblotting

Cells were lysed in 50 mM Tris pH 8.0, 150 mM NaCl, 0.5% NP-40, 50 mM NaF, 1 mM Na$_3$VO$_4$, 1 mM PMSF, one tablet protease inhibitors (EDTA free, Roche) per 10 ml, and 30 μg/ml DNaseI (Sigma-Aldrich) and analyzed by Western blot analysis. For detection of proteins by chemoluminescence (Advansta, K-12049-D50), a mouse anti-CHK1 (CS 2360, 2G1D5), rabbit anti-γH2A.X (CS 2577), rabbit anti-p53 (Santa Cruz sc-6243), rabbit anti-PARP1 (CS 9542), mouse anti-HSP90 (Santa Cruz sc-13119), mouse anti-CycD3 (BD 554195), mouse anti-hBCL2 (clone S100), rabbit anti-ATR-pSer428 (CS 2853), or rabbit anti-actin (CS 4967) was used. Goat anti-rabbit Ig/HRP (Dako, P0448) or rabbit anti-mouse Ig/HRP (Dako, P0161) was used as secondary reagent.

### Histology

For fixation and embedding E13.5, embryos were transferred to 4% paraformaldehyde (PFA in PBS) for fixation. Embryos were washed with 1 mM TBS (50 mM Tris-Cl, pH 7.5 150 mM NaCl) 2 × 15 min, 2 × 30 min, 70% EtOH, 80% EtOH, and 90% EtOH each for 30 min followed by 3 × 60 min incubation with 100% EtOH for

dehydration. To remove the ethanol, embryos were incubated in methyl benzoate for a minimum of 24 h. Embryos were incubated for 2 × 30 min with benzol, then with a 1:1 benzol-paraffin mixture at 60°C, and then in molten paraffin at 60°C. After positioning the embryos, paraffin dishes were placed on a cooling plate to enable a quick and homogenous hardening of the paraffin. Histology blocks were cut sagittal with a rotation microtome (Reichert.Jung, Leica 2040) and sections pressed on silanized glass slides.

### TUNEL assay

Slides were incubated 3 × 10 min with xylene, 3 × 5 min with 96% EtOH, 5 min with 70, 50, 30% EtOH, and PBS for rehydration. Slides were re-fixed with 4% PFA in PBS for 20 min, washed with PBS for 30 min, and permeabilized using 0.1% Triton X-100, 0.1% sodium citrate in a.d. for 2 min on ice. Slides were dehydrated after another washing step with TBS using 5-min incubation with 50, 70, and 96% EtOH, rinsed in chloroform, and then dried for 1 h at 37°C in a moist chamber. The TUNEL reaction was carried out by incubating sections with 15 μl of 0.6 μM FITC-labeled 12-dUTP (Roche, Basel, Switzerland), 60 μM dATP, 1 mM CaCl$_2$, and terminal deoxynucleotidyl transferase buffer (30 mM Tris, pH 7.2, 140 mM sodium cacodylate), and 25 units of terminal deoxynucleotidyl transferase (Roche, Basel, Switzerland) for 1 h at 37°C and covered with a plastic coverslip. The reaction was stopped by the addition of 10 mM Tris/1 mM EDTA, pH 8.0, for 5 min. Sections were washed twice in TBS before adding DAPI (2 μg/ml, in TBS, 40 μl/slide) for 10 min. Slides were washed twice with TBS and mounted in Mowiol® 40-88 (Sigma-Aldrich; 15 μl/slide).

### Statistical analysis

Statistical analysis was performed using unpaired Student's *t*-test or analysis of variance (ANOVA) for multiple group comparisons using Prism GraphPad software.

**Expanded View** for this article is available online.

### Acknowledgements

We are grateful to K. Rossi, C. Soratroi, I. Gaggl, M. Fischer, J. Vier, and J. Blitz for excellent technical assistance or animal care. We also thank J. Adams, S. Lowe, S. Elledge, and T. Mak for sharing mouse models. This work was supported by the FWF-funded Doctoral College "Molecular Cell Biology and Oncology" (W1101) and grant # I3271, "New insights into the BCL2 family" (FOR2036) and by the Deutsche Forschungsgemeinschaft (Grant HA2128/19-1). F. Schuler was supported by a Doc-fellowship from the Austrian Academy of Science (ÖAW) as well as intramural funding from the Medical University of Innsbruck (MUI-Start 2018-01-006).

### Author contributions

FS performed experiments, analyzed data, prepared manuscript and figures; SA performed experiments with CD34$^+$ HSPCs; ME supervised experiments with CD34$^+$ HSPCs; GH provided Hoxb8-FL cells and related expertise; CM performed histological analysis; and AV designed research, analyzed data, wrote paper, and conceived study.

### Conflict of interest

The authors declare that they have no conflict of interest.

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
