## [Review Process File · EMBO Reports]

Checkpoint kinase 1 is essential for fetal and adult hematopoiesis

Fabian Schuler, Sehar Afreen, Claudia Manzl, Georg Häcker, Miriam Erlacher and Andreas Villunger

Review timeline:

Submission date:	7 September 2018
Editorial Decision:	29 October 2018
Revision received:	18 March 2019
Editorial Decision:	26 April 2019
Revision received:	22 May 2019
Accepted:	24 May 2019

Editor: Achim Breiling

Transaction Report:

1st Editorial Decision

29 October 2018

Thank you for the submission of your research manuscript to EMBO reports. We have now received reports from the three referees that were asked to evaluate your study, which can be found at the end of this email.

As you will see, all referees think that the manuscript requires a major revision to allow publication in EMBO reports. All three referees have a number of concerns and/or suggestions to improve the manuscript, which we ask you to address in a revised manuscript. As the reports are below, I will not detail them here.

Given the constructive referee comments, I would like to invite you to revise your manuscript with the understanding that all referee concerns must be addressed in the revised manuscript and/or in a detailed point-by-point response. Acceptance of your manuscript will depend on a positive outcome of a second round of review. It is EMBO reports policy to allow a single round of revision only and acceptance or rejection of the manuscript will therefore depend on the completeness of your responses included in the next, final version of the manuscript.

Revised manuscripts should be submitted within three months of a request for revision; they will otherwise be treated as new submissions. Please contact us if a 3-months time frame is not sufficient for the revisions so that we can discuss the revisions further.

Supplementary/additional data: The Expanded View format, which will be displayed in the main HTML of the paper in a collapsible format, has replaced the Supplementary information. You can submit up to 5 images as Expanded View. Please follow the nomenclature Figure EV1, Figure EV2 etc. The figure legend for these should be included in the main manuscript document file in a section called Expanded View Figure Legends after the main Figure Legends section. Additional

Supplementary material should be supplied as a single pdf labeled Appendix. The Appendix includes a table of content on the first page, all figures and their legends. Please follow the nomenclature Appendix Figure Sx throughout the text and also label the figures according to this nomenclature.

For more details please refer to our guide to authors:
<http://embor.embopress.org/authorguide#manuscriptpreparation>

Important: All materials and methods should be included in the main manuscript file.

See also our guide for figure preparation:
http://www.embopress.org/sites/default/files/EMBOPress_Figure_Guidelines_061115.pdf

Regarding data quantification and statistics, can you please specify, where applicable, the number "n" for how many independent experiments (biological replicates) were performed, the bars and error bars (e.g. SEM, SD) and the test used to calculate p-values in the respective figure legends. Please provide statistical testing where applicable. See:
<http://embor.embopress.org/authorguide#statisticalanalysis>

Please also format the references according to EMBO reports style. See:
<http://embor.embopress.org/authorguide#referencesformat>

We now strongly encourage the publication of original source data with the aim of making primary data more accessible and transparent to the reader. The source data will be published in a separate source data file online along with the accepted manuscript and will be linked to the relevant figure. If you would like to use this opportunity, please submit the source data (for example scans of entire gels or blots, data points of graphs in an excel sheet, additional images, etc.) of your key experiments together with the revised manuscript. Please include size markers for scans of entire gels, label the scans with figure and panel number, and send one PDF file per figure.

- a complete author checklist, which you can download from our author guidelines (<http://embor.embopress.org/authorguide#revision>). Please insert page numbers in the checklist to indicate where the requested information can be found.
- a letter detailing your responses to the referee comments in Word format (.doc)
- a Microsoft Word file (.doc) of the revised manuscript text
- editable TIFF or EPS-formatted single figure files in high resolution (for main figures and EV figures)

I look forward to seeing a revised version of your manuscript when it is ready. Please let me know if you have questions or comments regarding the revision.

REFEREE REPORTS

Referee #1:

The manuscript by Villunger and colleagues is a comprehensive analysis of the effects of blocking CHK1 activity in hematopoiesis. Using cultured hematopoietic progenitor cells (human and mouse) treated with a variety of CHK1 inhibitors, constitutive hematopoietic deletion of Chk1 in fetal liver, and inducible hematopoietic deletion of Chk1 in adult mice, the authors convey the following central points:

1. They demonstrate that inhibition of CHK1 triggers cell cycle arrest, induction of DNA damage, and results in a BAX and BAK-dependent apoptosis by activating BIM along with either NOXA or PUMA.
2. Even when cell death is inhibited, progenitor cells treated with CHK1 inhibitors fail to rescue differentiation potential in a colony forming assay.
3. Loss of CHK1 in fetal liver results in defective hematopoiesis in part by triggering apoptosis and this triggers the hyper-proliferation of long-term HSC progenitors which are lost by exhaustion.

4. Blockade of cell death cannot rescue hematopoiesis in the Chk1-deleted fetal livers.
 5. Inducible deletion of Chk1 in the hematopoietic system is initially detectable, but over time the deleted cells are lost from the mice indicating a strong-selection pressure against loss of CHK1. Taken together these data clearly demonstrate the importance of CHK1 in hematopoietic development and I don't have any real criticisms of the work. However, I am not sure whether anyone would have predicted any other outcome. This seems to be a well performed, but completely predictable set of experiments.

One rationale that is put forward is that this study may imply that pharmacological CHK1 inhibition might be toxic to hematopoiesis. While this might be the case, it is also just as likely that the inhibition of the CHK1 inhibitors will be transient and incomplete and therefore not generally toxic. Therefore, I am not sure that this rationale for publication is particularly compelling.

Minor Points:

- Figure 7E is not called out in the text when the immunoblotting data are discussed.
- The introductory material should be abbreviated and focused on the topic at hand. Rather than reviewing hematopoiesis, I would focus on the role of DNA damage and cell cycle control in the hematopoiesis. This will help to frame the study better.

Referee #2:

Schuler et al investigate the effect of inhibition or deletion of the Checkpoint kinase 1, Chk1, in fetal and adult hematopoiesis. The results are interesting, the manuscript is generally well written and the findings well displayed, with robust differences in many cases. However, there are several notable exceptions to this that need to be rectified.

I am doubting all of the conclusions made with the VavCreERT2 approach. It is unclear that Chk1 is efficiently deleted; most likely it is not. It would be great if the mTmG reporter could be used as a reliable indicator of the efficiency of deletion, but it cannot. It is very clear that deletion efficiency varies widely between different loci, even using the same deleter mouse (Cre strain). The Rosa locus is one of the most efficiently deleted; since the deletion of the Tomato gene is very low (as low as 10%, page 12), one or both Chk1 alleles are likely retained in the majority of cells. The authors must check this specifically for the Chk1 allele, by analyzing the DNA, RNA and/or protein levels, if they want to make the claims they make. With the current data, their conclusions are not convincing. Second, they have not sufficiently documented the "rapid counter-selection" of cells retaining Chk1. If the authors want to make claims on the role of Chk1 in adult mice, they would need to perform these experiments (maybe best with Chk1f/- mice) with known and robust levels of deletion, and if they want to argue for counter-selection this must be quantified over time period. I would suggest to perform these experiments over a longer time period; maybe with increased tamoxifen or by delivering tamoxifen via injections. It would also be beneficial to wait until the effects of tamoxifen on hematopoiesis have normalized (tamoxifen alters the cell composition in the bone marrow).

Some of the cell cycle data are puzzling: why is there no difference between control and Chk1-deficient cells in Figure 6B, the DAPI stain? Why are there no subG1 cells in panel B, whereas there is a substantial subG1 population in panel C? Similar concern for Figure S3.

Please quantify all data and verify that all are n=3 or more independent experiments to substantiate claims: Figures 1B, 6A, 7E, 8E, 8F.

Please also temper the conclusions: clearly differentiate between the mechanistic insights you have substantial evidence for from the parts that are suggestive conclusions - it is fine to speculate, but it should be clear which conclusions are supported by data and which are speculations.

Specifics:

In the abstract, correct "definite" to "definitive" hematopoiesis.

The introduction is unnecessarily long. The first two paragraphs could be completely eliminated; if portions are left in, they need to be edited to provide a more nuanced, less dogmatic view: several issues that are stated with certainty are under debate.

P6: change "pluripotent" to "multipotent"

Figure 1A/p 6: are these data enough to conclude "caspase-dependent mitochondrial apoptosis"?

Figure 2A: please show the Annexin V analyses

Figure 3D: I am surprised by the lack of GEMM, G and GM colonies, at least for control (wild-type) cells. Please comment.

P7, last sentence is unclear

Figure 4E: were samples treated with ACK before analysis? Why or why not?

Figure 4F: cell frequencies are displayed, but the text on p 9 concludes "higher number". Please correct the text, but it may be valuable to also quantify the absolute numbers, in addition to frequencies

Figures 5E and F are inconsistent with each other: in E the fraction most enriched for HSCs (CD150+CD48-) is changes little, whereas in F there is a substantial decrease in HSCs (CD34-Flt3-). Similar for progenitor cells (CD48+ cells overlap substantially with Flt3+). This needs to be reconciled. [[compare to our own 150/48 profiles]] Enough events collected?

I am unsure of the authors' use of the term "stem cell exhaustion". On page 10, they refer to it - define it? - as loss of G0 cells. Is that consistent throughout? What exactly does "exhaustion" mean?

Figure 7A and B: are there significantly more LSK cells in p53^{-/-} mice? Are the phenotypes, or lack of phenotypes, in p53^{-/-} and Vav-Bcl2 mice consistent with previous reports?

It would be helpful to refer back to the figures in the discussion.

P14: "FL of these embryos"?

P14: the last sentence needs to be revised: colony formation ability is an experimental readout in vitro and does not account for lethality

P15: improve rationale of middle paragraph, as "MPP" are a major fraction of LSK cells.

P16: the sentence on "cycling cell types" and hepatocytes needs to be edited: hepatocytes can't both be a cycling cell type and proliferate minimally?

Please add statistics to Figure 1A

Figure S4 needs labeled y-axes

Referee #3:

In this study, Schuler and colleagues seek to evaluate the role of CHK1 in hematopoiesis. Using small-molecule inhibitors of CHK1, they first showed that CHK1 inhibition induced apoptosis in Hoxb8-immortalized progenitor cells and cord-blood derived CD34-positive cells and that this effect was significantly counteracted by either BCL2 overexpression or the use of QVD. They subsequently pursued their investigation by generating a conditional knockout of Chk1 in the hematopoietic compartment, thereby establishing that Chk1 is essential for the control of hematopoietic stem cell integrity and proliferation.

The work performed in this study is technically sound, very compelling, and thorough, with diverse in vivo (Chk1, Bcl2 and p53 knockout models) and ex vivo experiments using primary murine cells. Nonetheless, the concept that Chk1 is an important regulator of hematopoiesis is not entirely novel

because a study already reported that Chk1 haploinsufficiency resulted in anemia and dyserythropoiesis (Boles et al, Plos One, 2010).

Major comments:

- In figure 5A, it is somehow surprising that despite the exhaustion of the LSK compartment, we still observe a fair proportion of LK cells. Did these LK cell conserve their colony-forming capacity?

- In line with the notion that LK and LSK cells from Chk1^{fl/-} Vav-Cre⁺ mice are more prone to accumulate DNA damage compared to the WT cells (Figure 6), could the authors perform UV irradiation and/or etoposide treatment on those cells to confirm they are functionally more sensitive to DNA damage inducers?

- In addition to gH2AX staining, could the authors perform phospho-ATM and phospho-ATR staining on LK and LSK cells from WT and Chk1^{fl/-} Vav-Cre⁺ mice to confirm activation of those pathways in response to DNA damage.

- To confirm that p53 depletion and Bcl2 overexpression do not revert DNA damage accumulation in Chk1^{fl/-} Vav-Cre⁺ LK and LSK cells, could the authors complement their flow cytometry analysis in Figure 7A with a gH2AX staining? In line with this question, what compartment (LT-, ST-HSC, or MPP) within the LSK sub-population is the most affected by the accumulation of gH2AX?

- It is unclear why in Figure 8, the authors decided to develop a TAM-inducible conditionally deleted Chk1 mouse model to evaluate the effect of CHK1 inhibitors on established hematopoiesis? Would it not be more valuable to treat mice with available CHK1 inhibitors and characterize their effect on LK and LSK cells or more globally on the hematopoietic tree?

Minor comments:

- In figure 5A, the authors should depict in two different gates the KIT^{high} fraction equivalent to the control condition versus the KIT^{low} fraction only observed in the Chk1 KO condition. This will help the reader to understand better the difference in term of staining intensity between both conditions.

- A few typos remain in the manuscript: abstract "inhibtion", page 4 "protesomal".

1st Revision - authors' response

18 March 2019

Referee #1

The manuscript by Villunger and colleagues is a comprehensive analysis of the effects of blocking CHK1 activity in hematopoiesis. Using cultured hematopoietic progenitor cells (human and mouse) treated with a variety of CHK1 inhibitors, constitutive hematopoietic deletion of Chk1 in fetal liver, and inducible hematopoietic deletion of Chk1 in adult mice, the authors convey the following central points:

1. They demonstrate that inhibition of CHK1 triggers cell cycle arrest, induction of DNA damage, and results in a BAX and BAK-dependent apoptosis by activating BIM along with either NOXA or PUMA.
2. Even when cell death is inhibited, progenitor cells treated with CHK1 inhibitors fail to rescue differentiation potential in a colony forming assay.
3. Loss of CHK1 in fetal liver results in defective hematopoiesis in part by triggering apoptosis and this triggers the hyper-proliferation of long-term HSC progenitors which are lost by exhaustion.

4. Blockade of cell death cannot rescue hematopoiesis in the Chk1-deleted fetal livers.
5. Inducible deletion of Chk1 in the hematopoietic system is initially detectable, but over time the deleted cells are lost from the mice indicating a strong-selection pressure against loss of CHK1.

Taken together these data clearly demonstrate the importance of CHK1 in hematopoietic development and I don't have any real criticisms of the work. However, I am not sure whether anyone would have predicted any other outcome. This seems to be a well performed, but completely predictable set of experiments.

One rationale that is put forward is that this study may imply that pharmacological CHK1 inhibition might be toxic to hematopoiesis. While this might be the case, it is also just as likely that the inhibition of the CHK1 inhibitors will be transient and incomplete and therefore not generally toxic. Therefore, I am not sure that this rationale for publication is particularly compelling.

Response: This referee gives a concise summary of our work and acknowledges the technical quality of our experiments. This referee seems convinced that CHK1 ablation is detrimental to any cell type and hence the outcome predictable. While we will not try to change this reviewer's mind, but would like to point out that we all have learned on various occasions that certain things, predictable from published data, did not prove to be universally true when vigorously tested. Here, we see that initial observations made in cell lines and the early embryo appear to hold true in HSC of adult as well as embryonic origin, human or mouse derived. While this may be anticipated, nobody has thoroughly demonstrated these facts.

While this may be anticipated in the eyes of the referee, I can only point out that no one has documented this before, neither **(i)** during embryogenesis in the fetal liver, nor **(ii)** in adult hematopoiesis, nor **(iii)** for human CD34+ HSPCs. **(iv)** No one has explored cell death mediators engaged downstream of CHK1 within the BCL2 network, nor the interconnection with cell cycle arrest as default when death is blocked. As such, one can have a different opinion on the degree of "novelty" and "predictability" of our work. For example, we observed in our previous work on the role of CHK1 in B cells that mature resting B cells are still susceptible to CHK1 inhibition, despite the fact that these cells are resting, non-cycling (1). So, not all responses seem predictable based on the acknowledged role of CHK1 in cell cycle or the DNA-damage response. DT40 lymphoma cells can proliferate without CHK1, which no one would have predicted (2).

One of our arguments why these findings are of interest was indeed that clinical trials are exploring the suitability of CHK1 inhibitors for cancer treatment and, based on our results, we point out a potential *caveat* of such a strategy. Indeed, we can conclude that CHK1 inhibition, when achieved effectively, may cause side effects in the hematopoietic system (and presumably the GI tract, based on findings by others, (3)). Clearly, it is a question of the dosing regimen and time such inhibitors are applied to patients that will reveal if there is a therapeutic window for such drugs – clinical trials are ongoing. In fact, considering the result of our *in vivo* conditional deletion experiments in Figure 8, one can conclude that, toxic or not (a matter of dose), the hematopoietic system will adapt to CHK1 inhibition, one way or the other, potentially by counter-selection of cells resistant to the drug, but possibly also by maintaining HSPCs that may have suffered from insufficient CHK1 function and might show increased mutational load. With the emergence of age-related clonal hematopoiesis this might foster secondary malignancies. It is equally plausible that cancer stem cells that survive such treatment may harbor additional disease promoting mutations. Having said all this, we clearly do not have experimental evidence, nor are we in the position to conduct studies on secondary malignancies (beyond scope), nor can we perform dose escalation inhibitor studies *in vivo*, as already conducted by others, to document toxicity vs. anti-tumor activity. General toxicity of inhibitors can be ruled out, based on published data (4, 5), although detailed examination of normal hematopoiesis was not conducted in these studies.

Minor Points:

- Figure 7E is not called out in the text when the immunoblotting data are discussed. Thank you for noticing this error. Now, the figure is called out.

- The introductory material should be abbreviated and focused on the topic at hand. Rather than reviewing hematopoiesis, I would focus on the role of DNA damage and cell cycle control in the hematopoiesis. This will help to frame the study better.

We have substantially shortened the introduction putting an emphasis on DNA damage and CHK1 together with information of what is known about CHK1 in hematopoiesis.

Referee #2

Schuler et al investigate the effect of inhibition or deletion of the Checkpoint kinase 1, Chk1, in fetal and adult hematopoiesis. The results are interesting, the manuscript is generally well written and the findings well displayed, with robust differences in many cases. However, there are several notable exceptions to this that need to be rectified.

I am doubting all of the conclusions made with the VavCreERT2 approach. It is unclear that Chk1 is efficiently deleted; most likely it is not. It would be great if the mTmG reporter could be used as a reliable indicator of the efficiency of deletion, but it cannot. It is very clear that deletion efficiency varies widely between different loci, even using the same deleter mouse (Cre strain). The Rosa locus is one of the most efficiently deleted; since the deletion of the Tomato gene is very low (as low as 10%, page 12), one or both Chk1 alleles are likely retained in the majority of cells. The authors must check this specifically for the Chk1 allele, by analyzing the DNA, RNA and/or protein levels, if they want to make the claims they make. With the current data, their conclusions are not convincing.

Response: This referee is concerned that the reporter located in the easily accessible and open *Rosa26* locus that we used to trace deletion of *Chk1* may not faithfully read out target gene deletion, as it may simply be more easily accessible, compared to the *Chk1* locus. This would imply that *Chk1* may have never been deleted efficiently in GFP+ cells and not necessarily that only a few HSCs that retain CHK1 transcript and protein expression take over. This is a valid concern and similar phenomena have been noted by others (6).

We now provide new experimental evidence that in principle the mTmG reporter reliably reads out target gene deletion, at least when tested in vitro. We have generated Hoxb8-FL cells from either *Chk1^{fl/fl} mTmG* or *mTmG* mice. These cells were transduced with a retrovirus expressing CRE from the MSCV promoter. Cells turn green upon CRE expression while those that do not express CRE or fail to delete remain positive for dTomato. Cells were sorted based on fluorescence marker expression 48h after transduction. CHK1 protein expression was undetectable in GFP+ cells by western, which coincided with increased DNA damage, as indicated by increased gH2A.X levels. GFP+ cells were rapidly lost over time when also carrying the *Chk1^{fl/fl}* allele (Figure 8A). This suggests that these cells are outcompeted by Tomato+ cells, because they either stop cycling or undergo cell death. Similar results were obtained when a retrovirus encoding a CRE-IRES-GFP cassette was used in *Chk1^{fl/fl}* cells. GFP+ cells were sorted and analyzed by western 72h after transduction, confirming loss of CHK1 protein and increase DNA damage (Figure 8B). PCR analysis, however, still detected the floxed allele in DNA isolated from GFP+ cells (not shown), suggesting that some cells retain a functional CHK1 allele. We conclude that the mTmG allele allows faithful tracing of target gene deletion in the majority of Hoxb8-FL cells in vitro, but that deletion is incomplete. Our in vivo results further suggest that loss of CHK1 is not compatible with normal hematopoiesis selecting for non-deleting cells despite presence of active CRE. This is discussed now in more detail on page 15.

Second, they have not sufficiently documented the "rapid counter-selection" of cells retaining Chk1. If the authors want to make claims on the role of Chk1 in adult mice, they would need to perform these experiments (maybe best with *Chk1^{f/-}* mice) with known and robust levels of deletion, and if they want to argue for counter-selection this must be quantified over time period. I would suggest to perform these experiments over a longer time period; maybe with increased tamoxifen or by delivering tamoxifen via injections. It would also be beneficial to wait until the effects of tamoxifen on hematopoiesis have normalized (tamoxifen alters the cell

composition in the bone marrow).

Response: We fully agree with this referee and therefore we initially conducted western blot analysis of FACS-sorted bone marrow cells and thymocytes based on GFP vs. TOMATO reporter expression. We show that GFP+ cells that must have expressed CRE recombinase do still express CHK1. This referee is right in his/her critique, however, that our study currently lacks longitudinal data. So, we currently cannot discriminate if the residual CHK1 expression found in total bone marrow is due to counter-selection or incomplete deletion of *Chk1* on a per cell basis (e.g. cells may be heterozygous). Unfortunately, our breeding efforts were not successful and we do not have mice of the needed genotypes available in sufficient number for proper longitudinal analyses. As such, we are unable to deliver such experiments in a meaningful time frame. Based on this, and our findings made in cell lines (above), we have rephrased our wording in the results section and point out the possibility that HSCs or their immediate progeny may not have deleted CHK1 on both alleles.

Some of the cell cycle data are puzzling: why is there no difference between control and Chk1-deficient cells in Figure 6B, the DAPI stain? Why are there no subG1 cells in panel B, whereas there is a substantial subG1 population in panel C? Similar concern for Figure S3.

Response: The reason why there is no subG1 fraction in these FACS plots is explained by the fact that the analysis was performed on FACS-sorted, DAPI-negative (hence viable), LINnegative, LK or LSK cells. These cells were immediately fixed in EtOH after sorting. This is now pointed out more clearly on page 9 of the revised manuscript.

Please quantify all data and verify that all are n=3 or more independent experiments to substantiate claims: Figures 1B, 6A, 7E, 8E, 8F.

Response: We have increased the numbers of all our experiments to N=3, or higher, wherever feasible.

Ad Figure 1) we have now performed a time course analysis for *p21* mRNA induction using samples from three independent experiments and performed statistical analysis by ANOVA. Western blots have been repeated, yet, Figure 1C still shows our best blot.

Ad Figure 6) we have performed TUNEL staining originally on sections of two wt and two *Chk1* mutant embryos, but only showed one per genotype to demonstrate that there are substantially more TUNEL+ cell in the mutant embryos. We show only one representative IF-image, but now have quantified 3-4 independent fields per embryo section and genotype (i.e. 6-8 fields from two embryos/genotype) and quantified the TUNEL+ area/field using Image J (new version of Fig. 6A), as published before (7). To back up our findings, we now performed western blot analysis using total fetal liver extracts from three individual embryos lacking Chk1 (*Chk1*^{fl/fl} Vav-Cre), one heterozygote (*Chk1*^{fl/+} Vav-Cre) and two wild type controls (*Chk1*^{fl/+} Vav-Cre). Antibodies specific for activated ATR, ATR_S428 were used to corroborate our results (Figure 6B). Together with our flow-cytometric analysis of increased levels of gH2A.X in sorted LK/LSK cells from such embryos, we believe, we have made a convincing case that there is increased DNA damage and activation of the ATR checkpoint.

Ad Figure 7) we actually do show three animals per genotype in the western analysis provided in Figure 7E, which we hope will be deemed sufficient.

Figure 8) we agree, Figure 8E looks somewhat inferior, but we actually pooled bone marrow cells, FACS-sorted from three!! TAM-treated mice per lane to get enough cells and protein to run a western analysis. Figure 8F contains cells from thymi from three individual *Chk1* mutant TAM-treated animals, sorted on the basis of mGmT reporter expression, plus the relevant controls, which we hope will be seen as sufficient.

Please also temper the conclusions: clearly differentiate between the mechanistic insights you have substantial evidence for from the parts that are suggestive conclusions - it is fine to speculate, but it should be clear which conclusions are supported by data and which are speculations.

As suggested by this referee, we have adjusted the conclusions on counter selection in vivo, as admittedly our data is currently not strong enough (end of page 12/13).

Specifics:

In the abstract, correct "definite" to "definitive" hematopoiesis.

Corrected

The introduction is unnecessarily long. The first two paragraphs could be completely eliminated; if portions are left in, they need to be edited to provide a more nuanced, less dogmatic view: several issues that are stated with certainty are under debate.

We shortened the introduction substantially, focusing on the role of CHK1 and the limited information available on its role in hematopoiesis.

P6: change "pluripotent" to "multipotent"

Corrected

Figure 1A/p 6: are these data enough to conclude "caspase-dependent mitochondrial apoptosis"?

We have now treated HOXB8-FLT3 cells with CHK1 inhibitors ± the pan-caspase inhibitor qVD, showing that this prevents cell death at the time of analysis (new Figure 1B)

Figure 2A: please show the Annexin V analyses

The corresponding Annexin V analysis is now shown in Figure S2

Figure 3D: I am surprised by the lack of GEMM, G and GM colonies, at least for control (wildtype) cells. Please comment.

We have consulted our partners in Freiburg, conducting the human work. Apparently, it is common that very few such colonies form in vitro from human CD34+ cord blood (8). We only had one or two of those colonies in the dish. For simplification, we now pooled the few colonies and refer to "myeloid colonies" in Figure 3D and S3B.

P7, last sentence is unclear

We have rephrased this sentence to point out that LSK cells sorted from E13.5 embryos to put in culture may be less sensitive to CHK1i, as they are proliferating less at that time in development and are also extracted from their proliferation-permissive environment.

Figure 4E: were samples treated with ACK before analysis? Why or why not?

We have not treated the samples with ACK (red blood cell lysis buffer) to allow faithfully analysis of all erythroid stages in the fetal liver.

Figure 4F: cell frequencies are displayed, but the text on p 9 concludes "higher number".

Please correct the text, but it may be valuable to also quantify the absolute numbers, in addition to frequencies

We now only show absolute numbers in Figure 4 and moved frequencies to the supplement to avoid confusion. The text has been corrected accordingly.

Figures 5E and F are inconsistent with each other: in E the fraction most enriched for HSCs (CD150+CD48-) is changes little, whereas in F there is a substantial decrease in HSCs (CD34-Flt3-). Similar for progenitor cells (CD48+ cells overlap substantially with Flt3+). This needs to be reconciled. [[compare to your own 150/48 profiles]] Enough events collected?

This referee points out the mentioned inconsistency of our results using different cell surface marker panels to define HSC and MPP, both frequently used in the literature. We have now re-analyzed the data and came to the conclusion that the number of events collected for in our CD150/CD48 analysis may not have been sufficient to yield reliable results. Hence, we decided to only show results using FLT3/CD34 throughout the manuscript.

I am unsure of the authors' use of the term "stem cell exhaustion". On page 10, they refer to it - define it? - as loss of G0 cells. Is that consistent throughout? What exactly does "exhaustion" mean?

We refer to exhaustion as loss of stemness based on repeated mobilization out of dormancy causing DNA damage, telomere erosion and loss of clonogenic potential, yet, we avoid that phrase and use stem cell loss instead. We hope we have clarified this point.

Figure 7A and B: are there significantly more LSK cells in p53^{-/-} mice? Are the phenotypes, or lack of phenotypes, in p53^{-/-} and Vav-Bcl2 mice consistent with previous reports?

As this reviewer spotted correctly, we see a significantly higher percentage of aberrant “cKit low” LSK cells in CHK1-deficient embryos that lack p53 or overexpress BCL2. It has been reported that overexpression of BCL2 can increase the number of HSC by a factor of two (9) although we did not observe such changes in Vav-BCL2 transgenic mice (8). Yet, we see a tendency of increased FL cellularity in Chk1^{fl/-} Vav-Cre mice that overexpress BCL2. However, this difference did not show statistical significance, as seen also in the absence of p53. Hence, a potential explanation here could be a transient delay in cell death of these cells (more pronounced in the presence of BCL2 compared to loss of p53), or a slower transition of these cells into the LK cell pool. No such analysis has been done before to our knowledge.

It would be helpful to refer back to the figures in the discussion.

We now refer to figures in the discussion.

P14: "FL of these embryos"?

Fetal liversof these embryos

P14: the last sentence needs to be revised: colony formation ability is an experimental readout in vitro and does not account for lethality

Corrected to: This DNA damage may trigger cell death of fetal liver resident LSK cells that are cKitlow accounting for the embryonic lethality seen in *Chk1^{fl/-} Vav-Cre* mice.

P15: improve rationale of middle paragraph, as "MPP" are a major fraction of LSK cells.

Corrected to: ... suggesting that loss of CHK1 dominantly affects the survival of actively cycling HSPCs.

P16: the sentence on "cycling cell types" and hepatocytes needs to be edited: hepatocytes can't both be a cycling cell type and proliferate minimally?

Corrected to: Collectively this suggests that deletion of CHK1 is lethal for cycling cell types, with the notable exception of chicken DT40 B lymphoma cells (2). Consistently, adult hepatocytes are unaffected by loss of CHK1, most likely as homeostatic hepatocyte proliferation is minimal (10, 11).

Please add statistics to Figure 1A

We have provided a supplementary figure where statistical differences in the fraction of subG1 and G1 cells are indicated (Figure S1A)

Figure S4 needs labeled y-axes

Corrected

Referee #3

In this study, Schuler and colleagues seek to evaluate the role of CHK1 in hematopoiesis. Using small-molecule inhibitors of CHK1, they first showed that CHK1 inhibition induced apoptosis in Hoxb8-immortalized progenitor cells and cord-blood derived CD34-positive cells and that this effect was significantly counteracted by either BCL2 overexpression or the use of QVD. They subsequently pursued their investigation by generating a conditional knockout of Chk1 in the hematopoietic compartment, thereby establishing that Chk1 is essential for the control of hematopoietic stem cell integrity and proliferation.

The work performed in this study is technically sound, very compelling, and thorough, with diverse in vivo (Chk1, Bcl2 and p53 knockout models) and ex vivo experiments using primary murine cells. Nonetheless, the concept that Chk1 is an important regulator of hematopoiesis is not entirely novel because a study already reported that Chk1 haploinsufficiency resulted in anemia and dyserythropoiesis (Boles et al, Plos One, 2010).

Response: We thank this reviewer for acknowledging the overall quality of our work. I would

like to point out that the study cited here by Boles and colleagues is cited by us but exclusively addresses the impact of *Chk1* haplo-insufficiency on erythropoiesis, reporting on anemia phenotypes with about 30% penetrance in aged *Chk1*^{+/-} mice. This study shows that reduced CHK1 levels can impact on erythropoiesis with age, potentially by interfering with enucleation of erythroblasts, but do not address the role of CHK1 in HSC or HSPC. It only shows that *Chk1* mRNA can be detected in HSC in Figure 1A using two animals, nothing else. As such, we think it is premature to downgrade the novelty of our findings based on observations made in 30% of *Chk1*^{+/-} mice showing defective erythropoiesis and conclude a general role of CHK1 in hematopoiesis based on this.

Major comments:

- In figure 5A, it is somehow surprising that despite the exhaustion of the LSK compartment, we still observe a fair proportion of LK cells. Did these LK cell conserve their colony-forming capacity?

Response: Our data strongly suggests that they do not, as total fetal liver cells, containing LK cells do not form colonies (Fig. 5D).

- In line with the notion that LK and LSK cells from *Chk1*^{fl/-} *Vav*-*Cre*⁺ mice are more prone to accumulate DNA damage compared to the WT cells (Figure 6), could the authors perform UV irradiation and/or etoposide treatment on those cells to confirm they are functionally more sensitive to DNA damage inducers?

Response: This reviewer raises the question if CHK1-KO LSK/LK cells are more prone to accumulate DNA damage than HSCs from wt mice in response to UV (a trigger of ssDNA breaks) or etoposide (inducing dsDNA breaks) and asks us to test this. We actually do show a substantial increase in the levels of DNA damage in freshly isolated fetal liver resident LK and LSK cells (Fig 6A,B,C). The main obstacle to test this is the amount of cells that can be isolated from such mutant embryos and put in culture for *in vitro* analyses and the already high background of DNA damage noted *in situ*, potentially hampering detection of additional DNA damage signals. Please note that we talk of less than 10.000 cells/embryo (Figure 5B) that we have isolated for immediate staining in Figure 6.

- In addition to gH2AX staining, could the authors perform phospho-ATM and phospho-ATR staining on LK and LSK cells from WT and *Chk1*^{fl/-} *Vav*-*Cre*⁺ mice to confirm activation of those pathways in response to DNA damage.

As pointed out above, given the low number of cells available for analysis, the best we could come up with is to show that we have an increase in ATR, phosphorylated on S428, indicative of activation of the ATR driven arm of the DNA-damage response pathway in these fetal liver cells when CHK1 is lacking (Figure 6B). We hope this will suffice to satisfy this referee.

- To confirm that p53 depletion and Bcl2 overexpression do not revert DNA damage accumulation in *Chk1*^{fl/-} *Vav*-*Cre*⁺ LK and LSK cells, could the authors complement their flow cytometry analysis in Figure 7A with a gH2AX staining? In line with this question, what compartment (LT-, ST-HSC, or MPP) within the LSK sub-population is the most affected by the accumulation of gH2AX?

This reviewer is asking us to generate embryos of wt, *p53*^{-/-} and BCL2 transgenic mice lacking CHK1 to test for signs of DNA damage in different stem cell populations (LT, ST or MPP) and to address if lack of p53 or overexpression of BCL2 affects the accumulation of DNA damage in the absence of CHK1. A prediction would be that cells with high BCL2 levels may even accumulate more DNA damage, before they die or arrest.

As we no longer maintain the *p53/Chk1*^{fl/fl}/*Vav*-*Cre* breedings we investigated DNA damage in *Vav*-*BCL2/Chk1*^{fl/fl}/*Vav*-*Cre* mice. We were, however, only able to explore this in LK and LSK cells, but not in LT-, ST-HSC vs. MPP for technical reasons. As predicted, BCL2 transgenic cells displayed an even higher percentage of gH2AX positive cells (Figure S5C). These results are mentioned on page 10 of the revised manuscript.

- It is unclear why in Figure 8, the authors decided to develop a TAM-inducible conditionally deleted *Chk1* mouse model to evaluate the effect of CHK1 inhibitors on established hematopoiesis? Would it not be more valuable to treat mice with available CHK1 inhibitors and

characterize their effect on LK and LSK cells or more globally on the hematopoietic tree?

Response: We see this referee's point but general toxicity of inhibitors can be ruled out, based on published data (4,5; both cited) at least at concentrations used in these mouse studies. Yet, a detailed examination of normal hematopoiesis was not conducted in these studies. We were interested to see the potential effects of full and long-term inhibition of CHK1. Our results suggest that this is not possible, as long as HSPCs and their cycling progenitors are hit, leading to counter selection of hematopoietic cells that arise from HSPCs that have escaped deletion of CHK1. This has now been discussed in more detail and different context on page 15.

Minor comments:

- In figure 5A, the authors should depict in two different gates the KIT^{high} fraction equivalent to the control condition versus the KIT^{low} fraction only observed in the Chk1 KO condition. This will help the reader to understand better the difference in term of staining intensity between both conditions.

Response: We appreciate this valuable suggestion and now display data in Figure 5A accordingly, anticipating that this will now be easier to follow.

- A few typos remain in the manuscript: abstract "inhibtion", page 4 "protesomal"
Done!

References:

- Schuler F, Weiss JG, Lindner SE, Lohmuller M, Herzog S, Spiegl SF, et al. Checkpoint kinase 1 is essential for normal B cell development and lymphomagenesis. *Nat Commun.* 2017;8(1):1697.
- Zachos G, Rainey MD, Gillespie DA. Chk1-deficient tumour cells are viable but exhibit multiple checkpoint and survival defects. *EMBO J.* 2003;22(3):713-23.
- Greenow KR, Clarke AR, Jones RH. Chk1 deficiency in the mouse small intestine results in p53-independent crypt death and subsequent intestinal compensation. *Oncogene.* 2009;28(11):1443-53.
- Di Tullio A, Rouault-Pierre K, Abarrategi A, Mian S, Grey W, Gribben J, et al. The combination of CHK1 inhibitor with G-CSF overrides cytarabine resistance in human acute myeloid leukemia. *Nat Commun.* 2017;8(1):1679.
- Ferrao PT, Bucezynska EP, Johnstone RW, McArthur GA. Efficacy of CHK inhibitors as single agents in MYC-driven lymphoma cells. *Oncogene.* 2012;31(13):1661-72.
- Becher B, Waisman A, Lu LF. Conditional Gene-Targeting in Mice: Problems and Solutions. *Immunity.* 2018;48(5):835-6.
- Schuler F, Baumgartner F, Klepsch V, Chamson M, Muller-Holzner E, Watson CJ, et al. The BH3-only protein BIM contributes to late-stage involution in the mouse mammary gland. *Cell Death Differ.* 2016;23(1):41-51.
- Labi V, Bertele D, Woess C, Tischner D, Bock FJ, Schwemmers S, et al. Haematopoietic stem cell survival and transplantation efficacy is limited by the BH3-only proteins Bim and Bmf. *EMBO Mol Med.* 2013;5(1):122-36.
- Domen J, Gandy KL, Weissman IL. Systemic overexpression of BCL-2 in the hematopoietic system protects transgenic mice from the consequences of lethal irradiation. *Blood.* 1998;91(7):2272-82.
- Fujiwara T, Bandi M, Nitta M, Ivanova EV, Bronson RT, Pellman D. Cytokinesis failure generating tetraploids promotes tumorigenesis in p53-null cells. *Nature.* 2005;437(7061):1043-7.
- Pandit SK, Westendorp B, de Bruin A. Physiological significance of polyploidization in mammalian cells. *Trends Cell Biol.* 2013;23(11):556-66.

2nd Editorial Decision

26 April 2019

Thank you for the submission of your revised manuscript to our editorial offices. We have now received the reports from two of the three referees that were asked to re-evaluate your study (you will find enclosed below). As you will see, both referees now support the publication of your manuscript in EMBO reports. Referee #2 did not respond to my repeated invitations to reassess the

manuscript. However, going through your point-by-point response, I conclude that his/her concerns and suggestions have been adequately addressed.

Before we can proceed with formal acceptance, I have the following editorial requests we ask you to address in a final revised manuscript:

- The title is currently too long and reads rather complicated. Please provide a simpler and shorter title (with not more than 100 characters including spaces).
- By journal policy, we do not allow 'data not shown' (page 8 and page 11). Thus, please show these data either in the main or EV figures, or in an Appendix (or remove the phrase, if these data are not central to the study). See also: <http://embor.embopress.org/authorguide#unpublisheddata>
- Supplementary/additional data: By journal policy, you can submit up to 5 images as Expanded View. Currently, there are 7 EV figures (with figure 6 in two separate files - a figure can not be bigger than 1 page). Please select 5 figures as EV figures, and provide the remaining data (and maybe the 'data not shown' - see above) as a single pdf labeled Appendix. The Appendix needs page numbers, a table of content on the first page (with page numbers), and the figures and their legends. Please follow the nomenclature Appendix Figure Sx throughout the text and also label the figures according to this nomenclature. Please also change the callouts of the EV figures to Figure EVx throughout the manuscript text (presently, these are called out as Figure Sx), and uniformly use Figure EVx also in EV figure legends.

For more details please refer to our guide to authors:

<http://embor.embopress.org/authorguide#manuscriptpreparation>

See also our guide for figure preparation:

http://www.embopress.org/sites/default/files/EMBOPress_Figure_Guidelines_061115.pdf

- The labelling of the x-axis in Fig. 1A is too small (and rather illegible). Please provide this panel with bigger fonts.
- Fig EV1B is out of focus and the text is illegible. Please provide this in better quality and with bigger fonts.
- In general, the labelling of the y- and x-axes, and also of most of the text inside the FACS diagrams is too small (and rather illegible). Please provide all these panels with bigger fonts (Figs. 5A/C, 6C/D, 7A, EV/S2A/B, EV/S3B, EV/S4A, EV/S5C, EV/S7). Maybe, it would be a good idea to show the large FACS panels in more detail and increased size in the Appendix (e.g. EV1B, EV4A and EV7)?
- Please indicate in field D-10 in the author checklist (or in the text on page 18) that you comply with the ARRIVE guidelines. See also: <http://embor.embopress.org/authorguide#livingorganisms>
- Some of the WB panels (e.g. in Fig. 1C) have rather different contrast or brightness. Please show the WBs with panels with equal contrast, and as unmodified as possible.
- As they are significantly cropped, could you provide the source data for the entire Western blots shown in the manuscript (including the EV figures)? The source data will be published in separate source data files online along with the accepted manuscript and will be linked to the relevant figures. Please submit scans of entire gels or blots together with the revised manuscript. Please include size markers for scans of entire gels, label the scans with figure and panel number, and send one PDF file per figure.
- Presently, for many bar diagrams conditions are shown that seem to lack statistical testing (e.g. Fig. 1E PF 500 nM). If these were conditions where no significant difference compared to the control has been observed, please indicate this (as in Fig. 3D - n.s.). Please go through all the figures and make sure that statistical testing has been performed for all conditions in all bar diagrams, and that this is indicated in the diagrams. Could statistical testing also be provided for Fig. 5B?

- Please provide scale bars for all microscopic images (e.g. Figs. 4A,C,D and 6A). Please do not write on the scale bars. Please indicate the size only in the respective figure legend.
- Finally, please find attached a word file of the manuscript text (provided by our publisher) with changes we ask you to include in your final manuscript text, and some queries (comments), we ask you to address. Please provide your final manuscript file with track changes, in order that we can see the modifications done.

- a Microsoft Word file (.doc) of the revised manuscript text
- editable TIFF or EPS-formatted figure files (main figures and EV figures) in high resolution (of those with changes)
- The Appendix file

In addition I would need from you:

- a short, two-sentence summary of the manuscript
- two to three bullet points highlighting the key findings of your study
- a schematic summary figure (in jpeg or tiff format with the exact width of 550 pixels and a height of about 400 pixels) that can be used as visual synopsis on our website.

I look forward to seeing the final revised version of your manuscript when it is ready. Please let me know if you have questions or comments regarding the revision.

REFEREE REPORTS

Referee #1:

After reading the revised version of the manuscript by Villunger and colleagues, I find the revisions provided acceptable. My chief concerns of the previous version were that these findings are predictable. While this concern is not completely alleviated, I find the authors' logic that while predictable, not one has performed such in depth experimentation. Therefore, I agree that the concepts presented will be of general interest and are worthy of report.

Referee #3:

After this first round of review, the authors addressed satisfactorily all points raised by the reviewer. This article is now suitable for publication in EMBO reports.

2nd Revision - authors' response

22 May 2019

The authors performed all minor editorial changes.

Corresponding Author Name: Andreas Villunger
Journal Submitted to: EMBO Reports
Manuscript Number: EMBOR-2018-47026V3